Implications of differences between recent anthropogenic aerosol emission inventories
on diagnosed AOD and radiative forcing from 1990 to 2019
Marianne T. Lund[1,*], Gunnar Myhre[1], Ragnhild B. Skeie[1], Bjørn H. Samset[1], Zbigniew Klimont[2]
1 CICERO Center for International Climate Research, Oslo, Norway
2 International Institute for Applied Systems Analysis (IIASA), Laxenburg, Austria
*Corresponding author: m.t.lund@cicero.oslo.no

## 9   Abstract

*This study focuses on implications of differences between recent global emissions inventories for*
*simulated trends in anthropogenic aerosol abundances and radiative forcing (RF) over the 1990-2019*
*period. We use the ECLIPSE version 6 (ECLv6) and Community Emission Data System year 2021*
*release (CEDS21) as input to the chemical transport model OsloCTM3 and compare the resulting*
*aerosol evolution to corresponding results derived with the first CEDS release, as well as to observed*
*trends in regional and global aerosol optical depth (AOD). Using CEDS21 and ECLv6 results in 3%*
*and 6% lower global mean AOD compared to CEDS in 2014, primarily driven by differences over China*
*and India, where the area average AOD is up to 30% lower. These differences are considerably larger*
*than the satellite-derived interannual variability in AOD. A negative linear trend over 2005-2017 in*
*global AOD following changes in anthropogenic emissions is found with all three inventories but is*
*markedly stronger with CEDS21 and ECLv6. Furthermore, we confirm that the model better captures*
*the sign and strength of the observed AOD trend over China with CEDS21 and ECLv6 compared to*
*using CEDS, while the opposite is the case for South Asia. We estimate a net, global mean aerosol-*
*induced RF in 2014 relative to 1990 of 0.08 W m$^{-2}$ for CEDS21, and 0.12 W m$^{-2}$ for ECLv6, compared*
*to 0.03 W m$^{-2}$ with CEDS. Using CEDS21, we also estimate the RF in 2019 relative to 1990 to be 0.10*
*W m$^{-2}$, reflecting the continuing decreasing trend in aerosol loads post 2014. Our results facilitate more*
*rigorous comparison between existing and upcoming studies of climate and health effects of aerosols*
*using different emission inventories.*

## 29   1 Introduction

Human activities have led to a substantial increase in atmospheric abundances of aerosols relative to
pre-industrial conditions. While increasing emissions of greenhouse gases is the dominant driver of
recent global warming, aerosols play a key role in shaping regional and global climate, and for
anthropogenic climate change, through their interactions with radiation and clouds. The sixth assessment
report (AR6) of the Intergovernmental Panel on Climate Change (IPCC) estimates that changes in
atmospheric aerosols have contributed an effective radiative forcing (ERF) of –1.3 W m$^{-2}$ over the
industrial era (1750–2014), albeit with a wide uncertainty range of –2.0 to –0.6 W m$^{-2}$ (Forster et al.,
37   2021).

Over recent decades, anthropogenic emissions of aerosols and their precursor gases have changed
rapidly, with substantial spatiotemporal heterogeneity, particularly in Asia. Following decades of rapid
economic growth in China, the combustion of coal, other fossil fuels, and biofuels increased
considerably, resulting in the region becoming the dominant source of air pollution emissions. However,
since the adoption of the national action plans targeting particulate matter levels (i.e. Air Pollution

Prevention and Control in 2013 (SCPRC, 2013) and Winning the Blue Sky Defense Battle in 2018 (SCPRC, 2018)), emissions of sulfur dioxide ($SO_2$) and then nitrogen oxide (NOx) in China have declined rapidly (Klimont et al., 2017; Klimont et al., 2013; Tong et al., 2020; Zheng et al., 2018). Recent studies suggest that also black carbon (BC) emissions are declining (Kanaya et al., 2020; Zheng et al., 2018). A strong growth in emissions of $SO_2$ and other pollutants has been seen in South Asia (Kurokawa & Ohara, 2020), resulting, according to studies, in India overtaking China as the dominant emitter of $SO_2$ (Li et al., 2017). These contrasting trends have given rise to a distinct dipole pattern of increasing and declining aerosol optical depth over South and East Asia, respectively, visible in satellite data (Samset et al., 2019). Such rapid changes are likely to affect the climate of the regions, as aerosols have been shown to have a notable influence on regional temperature and precipitation, including extremes (e.g. Bollasina et al., 2011; Hegerl et al., 2019; Marvel et al., 2020; Samset et al., 2018; Sillmann et al., 2013), with different responses to scattering and absorbing aerosols. However, the exact nature and magnitude of such climate implications need to better quantified (Persad et al., 2022).

Robust quantification of the impacts of aerosols requires reliable and consistent estimates of anthropogenic emissions. However, currently there exist substantial differences, in both magnitudes and trends, between available emission inventories (e.g. Crippa et al., 2018; Elguindi et al., 2020; Smith et al., 2022). Emission inventories are quantifications of contributions from various industrial processes or other anthropogenic activities to the rate of emissions of various compounds to the atmosphere. They generally combine bottom-up information such as reported economic activities with direct observations and process modelling and are used extensively in essentially all efforts to quantify climate and air quality implications of human activities. While the overall scientific uncertainty on aerosol-induced global mean radiative forcing (RF) is larger than the estimated regional changes, the uncertainty also varies over the recent decades depending on the overall level of emissions and their location relative to cloud decks and other climate features (Bellouin et al., 2020; Regayre et al., 2014; Samset et al., 2019; Szopa et al., 2021). Hence, understanding both the inherent inventory differences and the implications of these on downstream calculations and modelled quantities such as aerosol optical depths and radiative forcing is crucial.

As an example, a critical issue that has recently been highlighted is a notable underestimation of the decline in Chinese emissions of $SO_2$ and NOx, and overestimation of carbonaceous aerosol emissions in Asia and Africa, in the Community Emission Data System (CEDS) developed for the sixth cycle of the Coupled Model Intercomparison Project (CMIP6) (Szopa et al., 2021). Recent work has shown that results from the CMIP6 experiments fail to fully capture the observed recent trends in aerosol optical depth (AOD) in Asia (Cherian & Quaas, 2020; Ramachandran et al., 2022; Su et al., 2021; Wang et al., 2021), with the discrepancy largely attributed to the misrepresentation of emissions in the region in last decade of the historical CMIP6 period. Other studies demonstrate that the poor representation of observed aerosol trends can propagate to further uncertainties in attribution of aerosol-induced impacts, such as the East Asian monsoon (Wang et al., 2022) and health impacts (Cheng et al., 2021). In addition to CMIP6, the CEDS emissions have also been used in individual model studies of historical aerosol evolution, radiative forcing, sector attribution, and air quality assessments (e.g. Bauer et al., 2020; Chowdhury et al., 2022; Lund et al., 2018; Lund et al., 2020; Paulot et al., 2018). Moreover, uncertainties and biases in the baseline historical inventory may influence scenario-based assessments of near-term future regional climate risk.

Since the initial parts of the CMIP6 exercise, the CEDS inventory has undergone several revisions. The most recent version from 2021, covering the period up to 2019, exhibit several key differences compared to the initial release – for some species all the way back to the early 2000s. More specifically, emissions of BC, OC and NOx are all substantially lower in the update, in global totals and, particularly, in Asia, and the decreasing trend in Chinese $SO_2$ is more pronounced. However, the implications of these differences in input data on simulated anthropogenic aerosol distributions, globally and regionally, and the resulting radiative forcing, have not been fully quantified and cannot be directly extrapolated.

Furthermore, as the update to CEDS came too late for uptake in IPCC AR6, it is pertinent to ask if the
influence of these emission inventory differences affected the assessed evolution of atmospheric aerosol
trends and subsequent climate implications.
Here, we present an investigation of the implications of known differences in recent emission inventories
on quantified aerosol burdens, optical depth, and radiative forcing, over the period 1990-2019. Using
the chemical transport model OsloCTM3, we perform simulations with the CEDS21 emission inventory
and compare to previously published results derived with the original CEDS release (Lund et al., 2018;
Lund et al., 2019). We also perform simulations with a third recent global inventory, the ECLIPSE
version 6b, where emissions are similar in evolution but generally even lower than in CEDS21,
especially in the most recent period. We explore the differences in simulated evolution of global and
regional anthropogenic aerosol loads between experiments using the different inventories, comparing
optical depth to remote sensing observations, and quantify the resulting radiative forcing. Our aims are
to document the model ability to represent recent observed aerosol trends and to quantify the
implications of differences in inventories available for the community on downstream diagnosed
quantities critical for assessing the air quality and climate implications of anthropogenic aerosol.

## 2 Methods


Atmospheric concentrations of aerosols are simulated with the global chemical transport model
OsloCTM3 (Lund et al., 2018; Søvde et al., 2012). The model is driven by meteorological data from the
European Center for Medium Range Weather Forecast (ECMWF) OpenIFS model updated every 3
hours and is run in a 2.25°x2.25° horizontal resolution, with 60 vertical levels (the uppermost centered
at 0.1 hPa). OsloCTM3 treats tropospheric and stratospheric chemistry, as well as modules for
carbonaceous, secondary organic, sulfate, ammonium-nitrate, sea salt and dust aerosols. Aerosols are
scavenged by convective and large-scale rain (ice and liquid phase), with rainfall calculated from
ECMWF data for convective activity, cloud fraction, and rainfall. Dry deposition applies prescribed
deposition velocities for different land cover types. For further details we refer to Lund et al. (2018) and
Søvde et al. (2012).
The aerosol optical depth (AOD) and instantaneous top-of-atmosphere radiative forcing due to aerosol-
radiation interactions (RFari) is calculated offline using a multi-stream model with the discrete ordinate
method DISORT (Myhre et al., 2013; Stamnes et al., 1988). The same radiative transfer model is also
used to estimate the radiative forcing of aerosol-cloud interactions (RFaci) (earlier denoted the cloud
albedo effect or Twomey effect). To account for the change in cloud droplet concentration resulting
from anthropogenic aerosols, which alter the cloud effective radius and thus the optical properties of the
clouds, the approach from Quaas et al. (2006), is used. Briefly, this approach is based on a statistical
relationship between cloud droplet number concentrations and fine-mode AOD derived from satellite
data from the MODerate Resolution Imaging Spectroradiometer (MODIS).
Modeled AOD is compared with retrievals from the MODIS instrument on the Aqua satellite, which is
available for the period 2003-2020 (MOD08, 2018). We use the combined Dark Target and Deep Blue
AOD at 550nm, release MOD08_M3_V6.1, downloaded from the NASA Giovanni interface. MODIS-
Terra AOD is also available for the same period and is, for most years, around 10% lower than MODIS-
Aqua on global average. However, based on previous evaluation of the MODIS AOD and a reported
drift in the Terra data (Levy et al., 2010; Sherman et al., 2017), we choose to use MODIS-Aqua for the
model comparison in the current study. Temporal trends in simulated and observed AOD are estimated
on global-mean and grid point basis by linear least square fitting and defined as statistically significant
(from no trend) when the linear Pearsons correlation coefficient is significant at the 0.05 level. To

minimize the influence of individual years, e.g. with higher biomass burning influence, we calculate a set of trends with one and one year removed from the sample and then take the average of this set of coefficients. Interannual variability is estimated on a grid point basis as the standard deviation of the residual when subtracting a 10-year boxcar average (with mirrored data around the end points). We also compare modeled AOD with ground-based measurements from the AERONET (AErosol RObotic NETwork) (Holben et al., 1998) Version 3 Level 2.0 retrievals at 500 nm. The comparison uses all available data from all months and stations for a given year, with modeled AOD linearly interpolated to the latitude and longitude of each station.

Five different time series of simulated aerosol distributions covering the 1990-2019 period are included in this analysis, using three different emission inventories and either fixed or actual (i.e. corresponding to the emission year) meteorology. The fixed meteorology runs forms the basis for investigating differences in simulated anthropogenic aerosol and corresponding RF, while the latter is used in the comparison with observed AOD. Table 1 provides a summary of the experiments.

Two sets of fixed meteorology simulations are performed using anthropogenic emissions from CEDS version 2021 (O'Rourke et al., 2021) (hereafter *"CEDS21"*) and ECLIPSEv6b baseline (hereafter *"ECLv6"*) inventories. The ECLv6 emissions are developed with the Greenhouse Gas - Air Pollution Interactions and Synergies (GAINS) model (Amann et al., 2011). Version 6b (IIASA, 2022) consists of gridded aerosol and reactive gas emissions in 5-year intervals over the period 1990-2015, as well as emissions for 2008, 2009, 2014 and 2016. The Community Emission Data System (CEDS) inventory provides a gridded inventory of anthropogenic greenhouse gas, reactive gases and aerosols since 1750 (Hoesly et al., 2018). In the first release, the most recent year was 2014, while the 2021 release covers the period until 2019. Simulations are performed for 1990, 1995, 2000, 2005, 2010, 2014 and 2016 emissions, as well as years 2018 and 2019 for CEDS21. Results from the current study are compared with previously published results from simulations over 1990 to 2014 performed with the first release of the CEDS emissions (hereafter *"CEDS"*) (Lund et al., 2018) and three of the SSP scenarios (SSP1-1.9, SSP2-4.5, and SSP3-7.0) from 2015 to 2100 (here we use data for 2020 and 2030) (Lund et al., 2019). These three scenarios broadly span the range of aerosol and precursor emissions projected in the SSPs. Keeping in line with the experimental design in Lund et al. (2018), we use year 2010 meteorological data and each simulation is run for one year, with 6 months spin-up. All three time series uses biomass burning emissions from van Marle et al. (2017) from 1990 to 2014 and Global Fire Emissions Database version 4 (GFED4, Randerson et al. (2017)) thereafter. We note that van Marle et al. (2017) emissions are also based on GFED. Other natural emissions (dust and sea salt aerosols, precursor gases from the ocean, soil, and vegetation) are fixed at the year 2010 levels.

For the comparison with MODIS data, we use a timeseries of OsloCTM3 simulations with CEDS emissions and actual meteorology covering the period 1990-2017 (the last three years uses Shared Socioeconomic Pathways (SSP) 2-4.5 emissions (Fricko et al., 2017) linearly interpolated between 2015 and 2020) (hereafter *"CEDSmet"*). These simulations were originally performed for the phase III of the Aerocom project (e.g. Gliß et al., 2021). For the present study, we also produce an updated version covering the 2001-2017 period using CEDS version 2021 emissions (hereafter *"CEDS21met"*). While differences in emissions exist also in the years prior, we restrict the use of resources by only going back to the start of the MODIS record, covering the period when the differences are most pronounced. In these simulations, the other natural aerosol emissions also vary following the meteorological year.

## 3 Results and discussion

Here we first document the differences in simulated global and regional aerosol abundances and trends arising from the spread between emission inventories. We then investigate how AOD diagnosed from experiments using old and new emission estimates compare with observed AOD. Finally, we present updated estimates of radiative forcing relative to 1990.

## 3.1 Influence of emission inventory differences on simulated aerosol distributions

Figure 1 shows global, total emissions of $SO_2$, BC, OC, NOx, ammonium ($NH_3$) and non-methane volatile organic compound (NMVOC) over the 1990-2019 period in the inventories used here. The differences are particularly pronounced after 2005. Both ECLv6 and CEDS21 show substantially lower emissions of most species during this period, relative to CEDS. In 2014, the largest relative differences between CEDS21 and CEDS are in BC and OC emissions, where CEDS21 is 20-30% lower. For $SO_2$, NOx, and NMVOC, the corresponding number is approximately 10%. ECLv6 is generally lower than both CEDS inventories, particularly for $SO_2$ and NMVOC, by about 30%. While not used in this study, we also note that similar differences have also been found between CEDS and two other recent global inventories, the Emissions Database for Global Atmospheric Research (EDGAR) version 5 (Crippa et al., 2020) and Hemispheric Transport of Air Pollution (HTAP) version 3 (Crippa et al., 2022).

Important geographical distinctions underlie these global differences, as demonstrated in Fig. S1 for selected main source regions. While a comprehensive investigation of causes for the inventory differences is beyond the scope of the present study, and can be difficult due to the number of underlying assumptions, input data, and revisions, we discuss some key features here. All three inventories rely on the energy statistical data from International Energy Agency (IEA), however, there are differences in assumptions about emission rates, implementation of policies, and data on non-energy sources. The ECLv6 estimates include explicit representation of air quality policies, and their implementation efficiency, drawing on national information and, if not available, extrapolation of trends considering capacity replacement (e.g., new vehicles, newly build power plant capacity) and emission performance of these new technologies. The result is, among other things, estimated faster decline of $SO_2$ and NOx emissions from power and industry (in turn in total emissions) in China over recent years than in CEDS (Fig. S1a,d). This decline has been also confirmed in Zheng et al. (2018). CEDS21 made a correction to CEDS, mirroring the estimates in the GAINS model for ECLv6. In South Asia, dominated by India, ECLv6 and CEDS21 show a similar difference to CEDS emissions of $SO_2$ and NOx, representing use of updated emission characteristics for coal power plants. India has had a slower economic growth and less heavy industry than China. While some policies aimed at controlling NOx from transport has been introduced, the limited polices in the power and industry sector have resulted in increasing Indian $SO_2$ and NOx emissions, but the growth has been slower than that in China in the 2000's. For BC and OC (Fig. S1b,c), the largest inventory differences are found in East Asia, mainly China, owing to differences in estimates of emissions from coal use in industry, with ECLv6 applying the lowest emission factors, and from open burning of municipal waste. For the latter category, CEDS has originally relied on the rather high estimates of waste generation and share burned (using Wiedinmyer et al. (2014)), while ECLv6 used independently estimated generation rates (Gómez-Sanabria et al., 2022). The declining BC trends in East Asia, as shown in ECLv6 and CEDS21, have been supported by measurements (e.g. Kanaya et al., 2020). Estimates for some species, e.g., $NH_3$, are often based on very similar sources of information as, apart from in Europe and North America, these have received less attention from policy making and measurement (emissions) community. Consequently, estimates are similar across all inventories at the aggregated regional level (Fig. S1e). Aside from East and South Asia, the overall

temporal evolution is generally similar in the main source regions across inventories, although magnitudes can differ.

### 3.1.1 Global and regional aerosol burdens in 2014

The differences between inventories are substantial enough to influence simulated aerosol burdens (i.e. column integrated aerosol mass, in mg m$^{-2}$) at the global mean level. For 2014, i.e. the most recent common year for all three emission inventories, we estimate 4% and 8% lower global mean burdens of total BC when using CEDS21 and ECLv6 (6% and 11% if considering only aerosols only from fossil fuel and biofuel combustion), respectively, compared to CEDS (Table S1). For primary organic aerosol (POA), the corresponding numbers are 11% and 13% (30% and 40%), while global mean total sulfate burden is 8% and 15% lower with CEDS21 and ECLv6. Smaller reductions in the order of 3-4% are also seen in the global mean SOA burden. Biogenic VOC emissions, the main source of SOA, are the same in all simulations. However, the SOA abundance is affected by the lower emissions of anthropogenic VOCs in both CEDS21 and ECLv6 than in CEDS (Fig.1), as well as by lower amount of POA, which serve as substrates for SOA formation.

For all these aerosol species, the burden differences are consistently largest over East Asia, followed by South Asia, and larger for ECLv6 than for CEDS21. Figure 2 shows absolute regional mean burden (with corresponding relative changes given in Fig.S3). Regions considered are East Asia (EAS), South Asia (SAS), Sub-Saharan Africa (SAF), North America (NAM), South America (SAM), North Africa and the Middle East (NAF), Europe (EUR), Southeast Asia (SEA), and Russia (RBU) (see also Fig. S2). For EAS, the new simulated burden of BC and POA is 30-40% lower, depending on inventory, compared to simulations using CEDS, following 50-60% lower BC and OC emissions. The 40-50% lower SO$_2$ emissions translate to 20-30% lower regional sulfate burden in our simulations. A similar relationship between emission and burden differences are simulated for SAS, where the burdens of BC, POA, and sulfate are 6%, 27%, and 30% lower, respectively, in experiments with ECLv6 than with CEDS. Lower burdens of sulfate and POA are simulated for all other regions as well, and in particular over NAF with ECLv6. In some regions, like SAM, NAF, and SAF, the new inventories estimate 20-30% lower BC emissions than CEDS, however, due to the lower absolute magnitudes, the simulated burden differences are small compared to other aerosols. We note that regional burdens can be influenced by long-range transport and thus affected by inventory differences outside the main source region. We also note that we find differences in surface concentrations between simulations that are broadly similar to the burden changes. While beyond the scope of the present study, this may have implications for assessments of air pollution related health impacts.

The only species that is globally more abundant in simulations with the two new inventories is nitrate. There is considerable regional heterogeneity, where the burden is lower compared to the CEDS experiments in South Asia and on the US east coast but higher in the US Midwest, parts of Africa and South America, and, especially, over East Asia (Fig.2, Fig.S3). While absolute differences are small in many regions compared to other species, the net effect is nevertheless a 15 and 24% higher global mean nitrate burden with CEDS21 and ECLv6, respectively, compared to using CEDS emissions. Changes in the atmospheric nitrate distribution result from a complex interplay between differences in emissions of NOx, NH$_3$, and SO$_2$. Studies have also shown that nitrate formation can be influenced by background concentrations of VOCs (e.g. Womack et al., 2019) We find the largest absolute difference in nitrate in EAS and SAS, however, of opposite sign. In EAS, emissions of SO$_2$ and NOx are both lower in ECLv6 and CEDS21 than in CEDS, whereas NH$_3$ emissions are higher (Fig.1, Fig.S1). This results in lower chemical competition for available sulfate and, in turn, enhanced formation of nitrate aerosol. In SAS, SO$_2$, NOx, and NH$_3$ are all lower in the two new inventories than in CEDS, as is the nitrate burden. Differences in concentrations of VOCs in the simulations with different inventories is a further complicating factor. Studies have suggested that nitrate formation can be more sensitive to changes in

VOCs than NOx, however, this is highly site specific (Yang et al., 2022). Further delineating the role of
individual factors on nitrate differences would require simulations beyond what is available for the
current study. The potential for an increasing relative role of nitrate for air pollution and climate in a
world with concurrent declines in $SO_2$ and NOx emissions but little in $NH_3$ has also been discussed in
previous studies (e.g. Bauer et al., 2007; Bellouin et al., 2011; Zhai et al., 2021). However, while more
studies have focused on local air pollution impacts of nitrate, and associated mitigation strategies, nitrate
is still missing from many global climate models. Moreover, when included, the model diversity in
simulated distributions is large (Bian et al., 2017). Our results suggest that uncertainties in emissions
and choice of inventory can contribute to spread in simulated nitrate aerosols and confound the
comparison of conclusions across modeling studies. Moreover, the complexity of the nitrate response
demonstrates that the impact of inventory differences on simulated aerosols cannot be understood from
scaling with the changes in individual emissions but requires explicit modeling.
To place the range in estimates between simulations with different inventories into more context, we
compare the differences in simulated aerosol burdens in 2014 to the difference in burdens over the 5-
year period from 2014 to 2019 using CEDS21. Both globally and regionally, the spread in burdens
between simulations with different inventories and the 2014-2019 burden changes are of the same order
of magnitude. In other words, at least in this case, the changes resulting from inventory differences are
as large as those due to the recent overall change in anthropogenic emissions.
Combined, these burden differences translate to a 3% and 6% lower global, annual mean AOD with
CEDS21 and ECLv6, respectively, compared to CEDS in 2014 in our simulations. As expected, the
differences are most pronounced over China and India (Fig. S4), where we estimate 20% and 30% lower
regional mean AOD in 2014 using the two new emission inventories, respectively, compared to using
CEDS. For context, Fig. S4 also shows the interannual variability in AOD from MODIS-Aqua (see Sect.
2): In these regions the differences between inventories are markedly larger than what can be expected
from natural year-to-year variations.

**3.1.2 Global and regional AOD 1990-2019**

Next, we take a closer look at differences in the simulated temporal trend, focusing on total AOD. Figure
3 shows the global and regional mean AOD from 1990 to 2019. Also shown is the linear trend from
2005 to 2017 for each of the timeseries. This period overlaps with the availability of remotely sensed
AOD discussed in Sect. 3.1.3, as well as the period with the most pronounced inventory differences.
However, as there is a certain extent of inventory differences prior to 2005, we also provide
corresponding linear trends over the full 1990-2017 period in Table S2.
The simulated AOD is consistently lower when using CEDS21 and ECLv6 emissions compared to
CEDS over the full period studied, with increasing divergence over time, especially after 2005. We
estimate a significant (at the 0.05 level - see Sect. 2) negative linear trend in global mean AOD of -0.005
and -0.006 per decade in simulations with CEDS21 and ECLv6, respectively. This trend strengthens
when extended to 2019 based on simulations with CEDS21. A negative global trend is also found when
using the first CEDS release, however, it is smaller and not significant over the period 2005-2014.
Extending the timeseries to 2017 by assuming that emissions follow SSP2-4.5 after 2014 (see Sect. 2),
as in Fig. 3, the negative trends strengthens and switches to significant as per our definition, but it
remains weaker than for the other two inventories. Considering the full period, we estimate a significant
negative trend in simulations with CEDS21 and ECLv6, but no trend when using CEDS (Table S2).
This long-term decline in total AOD is primarily driven by the decline in sulfate AOD, following the
emission decline after introduction of air quality policies, first in the US and Europe, then in China, and
the collapse of the Soviet Union (e.g. Aas et al., 2019). Over the full period, we simulate increasing
trends in BC and nitrate AOD, significant at the 0.05 level, with all three inventories (not shown),

however, their contributions to total AOD are much smaller than that of sulfate. Robust evidence of a declining influence by aerosols on climate since 1990 was recently found from observables (Quaas et al., 2022). Our model simulations capture this overall trend, and the findings reinforce the role of changes in anthropogenic emission, particularly since 2005. Furthermore, we suggest that if using the original CEDS emissions, models may have failed to capture this trend. We note that biomass burning emissions also change over time in our simulations, but we do not find any significant trend in biomass aerosols (BC and POA) AOD on the global mean scale over this period. We do note that years of high biomass burning activity, such as 2019 where GFED4 emissions are 25% higher than in 2018, can lead to marked jumps in simulated AOD. We have limited possible influence of such years on the linear trend calculated (see Sect. 2).

Regionally, we simulate significant declining trends in AOD over 2005-2017 for EUR and NAM, with this trend extending back to 1990 (Table S2), as expected. This is also consistent with surface observations both AOD and atmospheric sulfur and in agreement with other models (Mortier et al., 2020; Aas et al., 2019), and we capture the decline regardless of which emission inventory is used. In both regions, and across simulations with all three scenarios, we find a decline in the AOD of BC, OA, and sulfate, but an increasing trend in nitrate AOD. Over RBU, we also simulate a decline a significant decline in area average AOD over the full 1990-2017 period, but a flatter evolution when considering only 2005-2017. However, the results are similar with all three scenarios also here. In parts of the RBU region, GFED4 shows an increase in emissions over the latter period, resulting in a positive trend in the AOD of biomass aerosols from 2005. On the African continent, we simulate negative, albeit weak, trend in AOD over the 2005-2017 period for SAF. In contrast, the trend over the full period is positive. Anthropogenic emissions in SAF have increased (Fig. S1), although less steeply than in Asia, and we find significant increases in the AOD of all the anthropogenic species with all inventories from 1990 to 2017. However, from 2005 onwards, there has been a decreasing trend in GFED4 emissions, following a reduction in the burnt area of savannas (Wu et al., 2021). Biomass burning aerosols contribute relatively more to total AOD here than in the northern hemisphere regions and hence impose a stronger effect on the area average trend. A similar pattern is seen for SAM, while for SEA, another biomass burning influenced region, we find less clear trends. While diagnosed trends in total AOD in these regions are mostly of similar sign across simulations with the three inventories, we find that the trend in sulfate AOD diverges between model runs using CEDS or CEDS21 (positive trend) and ECLv6 (negative trend) in SAF and NAF, pointing to a need to better understand the drivers of emission changes in these regions and homogenize between inventories. As expected, the key differences between simulations with different inventories arise over Asia. Simulations with both CEDS21 and ECLv6 show a significant decreasing trend in total AOD over EAS between 2005 and 2017. While a decline is found using CEDS, it is much weaker and not significant. Moreover, differences between inventories affect the sign of the simulated trend when considering the full period, owing primarily to the spread in estimated sulfate AOD. For SAS, we simulate a consistent positive trend, but ranging from 0.01 per decade with ECLv6 to 0.03 per decade with CEDS, with increasing divergence in AOD over time. Similar magnitude differences between the sets of experiments exist for the AOD of all anthropogenic aerosol in this region.

### 3.1.3 Comparison with observed AOD

To explore whether the model captures observed global and regional trends better with the CEDS21 emissions than with CEDS, we compare simulated AOD to MODIS-Aqua retrievals and ground-based AERONET measurement. For this evaluation, we also use simulations where the model is driven by meteorology for the respective years, referred to as CEDSmet and CEDS21met (see Sect. 2), for more realistic comparison with the observations. Using both these, we also estimate negative linear trends in simulated global mean AOD from 2005 to 2017, strengthening from -0.001 per decade in CEDSmet to -0.003 per decade in CEDS21met. These are, however, weaker than the trends derived from the fixed

meteorology simulations in Sect. 3.1.2 (Fig. 3) and not significant at the 0.05 level, demonstrating the
notable influence of variability in meteorology and natural aerosols, masking trends due to changes in
anthropogenic emissions. This influence is particularly visible for the area averaged AOD for SAF and
NAF, where the diagnosed trend is positive but non-significant in these simulations, in contrast to the
negative trend found in simulations with fixed meteorology above. The negative trend over SAM is also
not significant at the at the 0.05 level in these runs. For other focus regions, results are similar between
fixed and actual meteorology runs and significant trends arise over the natural variability.
Figure 4a shows the annual, global mean simulated AOD from 1990 to 2017 and the MODIS-Aqua
AOD from 2003 to 2019. Dashed lines show the linear 2005-2017 trends. Figures 4b-d show the spatially
explicit trends. We first note that the magnitude of simulated global mean AOD is lower than that
derived from MODIS-Aqua, by around 20%. However, the overall geographical pattern of the observed
AOD is captured by the model (Fig. S5). Furthermore, the AOD simulated by the OsloCTM3 is within,
although in the lower range, of the spread in AOD between the CMIP and AeroCom models (Vogel et
al., 2022). As also shown by Vogel et al. (2022), there can be a notable spread in AOD derived from
different satellite products. They found a 13% standard deviation range in global mean AOD between
eight satellite products, with MODIS retrievals in the upper end. Although again the lower range, the
OsloCTM3 AOD falls within the full range of the satellite-derived annual mean AOD. Overall, this
suggests a reasonable OsloCTM3 performance in terms of magnitude and distribution.
In terms of temporal evolution, MODIS-Aqua data indicates a very weak positive linear trend of 0.001
per decade in global mean AOD over the 2005-2017 period (0.004 per decade when extending the data
to 2019). We do not, however, find this trend to be significant. MODIS data is influenced by substantial
year-to-year variability, in particular after 2010, which was also pointed out by Vogel et al. (2022).
Regions of significant positive observed AOD trend include parts of the ocean in the southern
hemisphere (Fig. 4b). Here, sea salt aerosols could be causing the increase. However, Quaas et al. (2022)
recently showed that this positive trend is not clear in Multi-angle Imaging SpectroRadiometer (MISR)
data. While we are focused on the anthropogenically-influenced regions in the present analysis, we
briefly note that the magnitude of the trends over the southern hemisphere oceanic regions is also not
captured by the model (Fig. 4c-d). We also simulate weaker trends in the boreal regions of North
America and Russia, contributing to the model-observation difference.
Over the main anthropogenic emission sources regions, there are significant observed declines in AOD
over East Asia, US, and Europe (Fig. 4b). These trends have been confirmed by both ground based and
remote sensing observations of AOD and other variables (Gui et al., 2021; Moseid et al., 2020; Paulot
et al., 2018; Quaas et al., 2022). For NAM and EUR, we calculate an area average negative observed
trend of -0.006 and -0.009, respectively, from MODIS-Aqua. This is of the same sign but weaker than
the trend simulated with both emission inventories. For the latter, this contrast findings by Mortier et al.
(2020), where models in general were found to underestimate the observed decrease in AOD seen in
surface observations. Over EAS, where the influence of inventory differences is most pronounced, a
significant negative observed trend of -0.044 per decade is calculated. This is in very close agreement
with the -0.40 per decade AOD trend simulated with the CEDS21, while simulations with CEDS do not
show a significant trend. Hence, the model is clearly able to better represent observed trends with the
updated inventory. This is further confirmed in Fig. 5, where we show five-year average deviations from
the period 2003-2017 in both MODIS-Aqua and simulated AOD. Using CEDS21 results in marked
improvements compared to observed AOD trends over China, both for the first and most recent full 5-
year periods. However, the opposite tendency is found for AOD over SAS. Here observations suggest a
significant positive trend of 0.04 per decade. The diagnosed trends are also positive in simulations using
both inventories, but somewhat weaker, especially when switching from CEDS to CEDS21 (and even
more so when using EClv6 emissions - Fig.3). Figure 5 suggests that this discrepancy arises in the more
recent decade. Furthermore, simulated AOD, and underlying emissions, suggest a leveling off in recent
years, which is not seen from MODIS-Aqua. Whether this is due to inaccurate representation of the

evolution of anthropogenic emissions in the inventories or could be influenced by poor model representation of other aerosols such as dust from agricultural soils and urban areas (e.g. construction, non-exhaust transport emissions), is however not clear from this analysis. We note that the model underestimates the magnitude of AOD observed by MODIS-Aqua in both EAS and SAS. To the extent that the MODIS is accurate, this could support the latter. This type of dust is suggested to give an important contribution to the particulate matter load (e.g. Chen et al., 2019; Xia et al., 2022), but are stilling missing from many global models. Other contributing factors include the representation of processes related to aerosol transport and scavenging. Finally, we also note that the 5-year deviations in Fig. 5 show quite some variability over the Middle East, with both positive and negative deviations from the baseline period. While anthropogenic emissions in this region increase steadily over the period (by 13-40% depending on species) in the inventories used in the present study, the strong influence from dust emissions in this region likely dominates the temporal variability.

A previous OsloCTM3 study by Lund et al. (2018) found an improved agreement between year 2010 ground-based AERONET observations and model output, including over Asia, when switching from CMIP5 and ECLIPSEv5 emissions to CEDS, the latter having higher emissions. This seemingly contradicts expectations following the now-known biases in this first release of CEDS. Here we repeat the comparison with AERONET, but for the year 2014. Resulting scatter density plots are given in Fig. S6. The normalized mean bias (NMB) compared to AERONET ranges from -21 to -29% in the simulations with fixed and actual meteorology. We find higher bias and lower correlation when switching from the original CEDS release to CEDS21 and ECLv6. Hence, while the model is better able to represent observed recent aerosol trends over East Asia with newer emission inventories, these results point to other issues that may have been concealed by too high anthropogenic emissions. Dust and atmospheric processing, as discussed above, are again possible contributing factors.

## 3.2 Impact of inventory differences on estimated anthropogenic aerosol RF

Finally, we quantify the aerosol-induced RF from the three sets of experiments. Figure 6a shows the RFari, RFaci, and net aerosol radiative forcing (RFnet, RFari plus RFaci) relative to 1990 for the three sets of experiments. The net RF of changes in anthropogenic (and biomass burning) aerosol is positive since 1990, except for 1995 and 2005, where a small negative forcing is estimated. As shown in Fig. 1, global anthropogenic $SO_2$ emissions show a peak in 2005 and the biomass burning emissions are relatively high. This positive global mean net RF is determined mainly by the balance between a positive forcing over the northern extratropics, dominated by aerosol-radiation interactions, and a negative forcing over Asia and parts of South America and Africa with stronger contributions from aerosol-cloud interactions (Fig. S7).

In 2014, we estimate a global mean RFnet of 0.03 W $m^{-2}$ for CEDS, 0.08 W $m^{-2}$ for CEDS21, and 0.12 W $m^{-2}$ for ECLv6 relative to 1990, of which the RFari constitutes 0.07 W $m^{-2}$, 0.09 W $m^{-2}$ and 0.10 W $m^{-2}$, respectively. We note that our framework only captures the cloud albedo effect and not radiative effects of any changes in cloud lifetime that may arise through the influence of aerosols (i.e. we calculate RF, not ERF). Our RFari estimate using CEDS emissions is similar to the multi-model mean RFari of 0.05 W $m^{-2}$ derived for the 1990-2015 period using ECLIPSE version 5 emissions by Myhre et al. (2017). The same study estimated a model mean RFnet of 0.1 W $m^{-2}$, but with a significant intermodel spread, from close to zero to more than 0.2 W $m^{-2}$. This spread is larger than the difference between estimates with different inventories in the present analysis. Nevertheless, the differences in emissions between CEDS and CEDS21 (ECLv6) translates to a factor 3 (5) stronger RFnet in our calculations.

Figure 6b shows regional mean RF, including the balance between RFari and RFaci. Following the significant decline in AOD over EUR and NAM, the dominant contributions to positive RF are found here, followed by Russia. There is however little difference between simulations with three inventories.

In contrast, the net RF over EAS switches sign from negative in simulations with CEDS to positive when using CEDS21 or ECLv6 due to observed decline in emissions now captured. While negative in all three sets of experiments, the net RF over SAS is 40% (20%) weaker when ECLv6 (CEDS21) emissions are used compared to CEDS. This results from a 50% (20%) lower net area averaged AOD change between 1990 and 2014, compared to simulations with CEDS.

The CEDS21 inventory extends to 2019, compared to 2014 in CEDS. The global mean net RF over this five-year period is estimated to be 0.10 W m$^{-2}$, driven primarily by a further positive forcing over China in line with the continued decline in SO$_2$ emissions following implementation of measures targeting improved air quality. Over India, the forcing in 2019 relative to 2014 remains negative, but weaker than during the preceding period, while over Europe and western Russia, the RF is low suggesting little further recent emission changes. We note however that this is a short period and results should be interpreted with that in mind. Using a selection of the SSP scenarios, Lund et al. (2019) extended simulations from 2014 CEDS emissions and quantified the projected aerosol-induced RF. The orange hatched bars in Fig. 6 show the range in RFnet in 2020 and 2030 (relative to 1990) estimated with SSP1-1.9, SSP2-4.5 and SSP3-7.0 in that study. The RFnet in 2019 estimated with CEDS21 here is close to the lower end of the bar, i.e. the RFnet projected under SSP3-7.0. However, prior to this higher biomass burning year, there are indications that the RFnet from simulations with CEDS21 tracked closer to SSP2-4.5 or an even lower emission pathway.

The dipole pattern of aerosol changes, and resulting RF, over India versus China that can be seen in observations and is expected to impose regional climate impacts, was first highlighted by Samset et al. (2019). Using emissions from CEDS and SSP1-1.9, SSP2-4.5 and SSP3-7.0, combined with a radiative kernel approach, that study estimated a range of 2014-2030 aerosol (SO$_2$ and BC) net RF of -1.0 W m$^{-2}$ (SSP1-1.9) to 0.82 W m$^{-2}$ (SSP2-4.5) over India, and 0.06 W m$^{-2}$ (SSP2-4.5) to 1.10 W m$^{-2}$ (SSP3-7.0) over China. Part of this range can be attributed to poor knowledge of current, and hence also future, regional emissions (Samset et al. 2019). In the present study, we estimate regionally averaged RFnet in 2019 relative to 2014 of -0.09 W m$^{-2}$ and 0.22 W m$^{-2}$ over India and China, respectively. For China, this recent RFnet is about 20% of the previously estimated difference between high and low future aerosol emission scenarios in 2030 (SSP2-4.5 and SSP3-7.0). Missing or incorrectly captured past emission trends can therefore markedly affect assessments of projected near-term aerosol-induced climate impacts, as they depend on a well constrained starting point.

## 4 Conclusions

We have investigated the impact of differences between recent global emission inventories available for the aerosol and climate modeling community on simulated anthropogenic aerosol abundances, and associated radiative forcing, from 1990 to 2019. Simulations with the chemical transport model OsloCTM3 and the CEDS emission inventory, developed for the sixth cycle of the IPCC, has been compared with corresponding results using two newer inventories: The CEDS 2021 update (CEDS21) and the ECLIPSE version 6b (ECLv6). Our objective was to evaluate the model performance considering revisions to the emissions input data, partly done to correct known regional biases, and to investigate the implications of inventory differences on downstream diagnosed quantities critical for assessing the air quality and climate effects of anthropogenic aerosol.

We have found that, apart for nitrate, simulations with the CEDS21 (ECLv6) inventory give lower global mean aerosol burdens than corresponding runs with CEDS, ranging from 4% (6%) for BC to approx. 10% (15%) for sulfate and POA in 2014 (the most recent historical year common for all scenarios). Differences are consistently most pronounced over East Asia, followed by South Asia, where they are on the order of 30-60% depending on species and scenario. Differences in the underlying anthropogenic emissions arise from different assumptions about emission rates, data on non-energy sources, and,

importantly, representation of air quality policies and their implementation efficiency. In our model, the global mean fine mode nitrate burden is 15% (24%) higher with CEDS21 (ECLv6) relative to CEDS, but with regional heterogeneity in sign of the difference. Overall, we estimate 3% (6%) lower total AOD with CEDS21 (ECLv6), respectively, compared to CEDS in 2014. The difference reaches approx. 20% and 30% over East and South Asia.

Over East Asia, we diagnose a significant negative linear trend in total area averaged AOD from 2005 to 2017 of -0.03 per decade in simulations using the ECLv6 emissions. In contrast, we find no significant trend in corresponding experiments with CEDS. Importantly, we find that the model is better able to capture the trend observed by MODIS-Aqua with both new inventories. In all three sets of simulations, we estimate a significant positive linear AOD trend over South Asia. The simulated trend is, however, weaker than that derived from MODIS-Aqua and this gap increases when switching from CEDS to the CEDS21 and ECLv6 inventories. We also underestimate the magnitude of observed AOD in the region, at least compared to this specific satellite product. Recent emission trends are less well constrained by observations in India than e.g. in China. The extent to which the model-observation difference arises from the input of anthropogenic emissions or could be influenced by poor model representation of other aerosols sources or atmospheric processes, is not clear from the present analysis. For other regions considered, there is generally agreement in the sign of the simulated area averaged AOD trend between the three sets of simulations, although the magnitude can differ, in particular for the AOD of individual species. For instance, there is an increasing (over time) divergence in the sulfate AOD over Africa between simulations using CEDS and ECLv6. Over most regions, nitrate AOD increases, however, nitrate contributes relatively less to total AOD than sulfate and OA.

Using offline radiative transfer calculations, we estimate a global mean net aerosol RF in 2014 relative to 1990 of 0.03 W m$^{-2}$ for CEDS, 0.08 W m$^{-2}$ for CEDS21, and 0.12 W m$^{-2}$ for ECLv6. Regionally, the sign of the net aerosol-induced RF switched from negative to positive when replacing CEDS emissions with CEDS21 or ECLv6 in our study. Hence, the failure to capture recent observed emission trends in China may have resulted in the wrong sign in estimates of the regional effect on the energy balance over recent decades. Over South Asia, the area average net RF is up to 40% lower in simulations with the updated inventories compared to CEDS.

While the focus of the present study is on anthropogenic aerosols, our comparison with observed AOD reveals potential issues related to the representation of natural aerosols or other processes in the OsloCTM3. In particular, the model does not capture the strength of the positive AOD trend observed over high latitude North America and Russia, likely due to an increase in biomass burning aerosols. For individual years, we also find a larger underestimation in AOD compared to AERONET measurements when switching from CEDS to the lower CEDS21 and ECLv6 emissions, despite better representation of some key regional observed trends. Further studies are required to investigate this in more detail.

Anthropogenic aerosols are changing rapidly, particularly in Asia, with potentially large but insufficiently quantified implications for regional climate. We have demonstrated that differences between recent emission inventories translate to notable differences in global and regional trends in anthropogenic aerosol distributions, and in turn in estimates of radiative forcing. Although additional studies are required to fully quantify the broader implications for aerosol-induced climate and health impacts, our results facilitate comparisons between existing and upcoming studies, using different emission inventories, of anthropogenic aerosols and their effects.

**Code availability**

The OsloCTM3 is available on from https://github.com/NordicESMhub/OsloCTM3.

**Data availability**

Model data underlying the manuscript figures are available from 10.6084/m9.figshare.20254764.
AERONET data is downloaded from https://aeronet.gsfc.nasa.gov/, MODIS data from
https://giovanni.gsfc.nasa.gov/giovanni/, and CEDS21 emissions from the PNNL DataHub
https://doi.org/10.25584/PNNLDataHub/1779095.

**Acknowledgements**

This work has been conducted with support from the Research Council of Norway (grants 248834,
314997 and 324182). The authors acknowledge the UNINETT Sigma2 – the National Infrastructure
for High Performance Computing and Data Storage in Norway – resources (grant NN9188K).

**Author contributions**

MTL led the study design and analysis and the writing. The OsloCTM3 model experiments were
performed by MTL and RBS. GM performed the radiative transfer modeling and BHS contributed
graphics, silly jokes, and MODIS analysis. All authors contributed to the discussions and writing.

**Competing interests**

The authors declare that they have no conflict of interest.

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

**Tables:**


*Table 1: Summary of experiments used in the study.*

| Name | Description | Years simulated |
|---|---|---|
| CEDS | CEDS v2016 emissions, fixed meteorology | 1990, 1995, 2000, 2005, 2010 2014 |
| CEDS21 | CEDS v2021 emissions, fixed meteorology | 1990, 1995, 2000, 2005, 2010 2014, 2016, 2018, 2019 |
| ECLv6 | ECLIPSEv6b emissions, fixed meteorology | 1990, 1995, 2000, 2005, 2010 2014, 2016 |
| CEDSmet | CEDS v2017 emissions until 2014 and SSP2-4.5 for 2015-2017, running meteorology | 1990-2017 |
| CEDS21met | CEDS v2021 emissions, running meteorology | 2001-2017 |
























**Figures:**

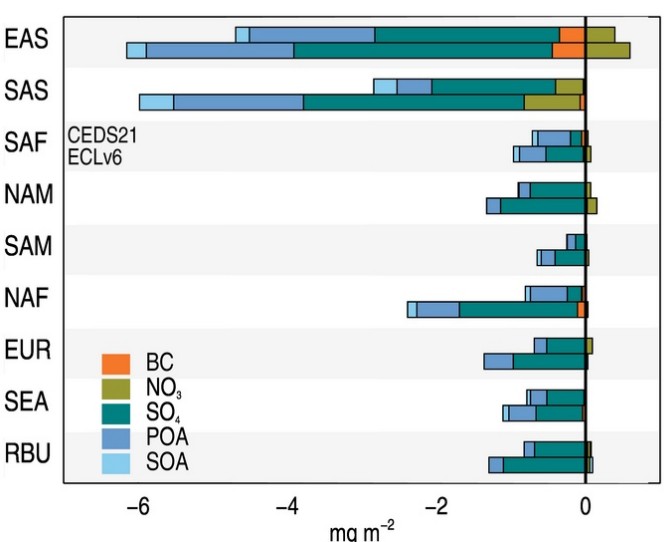


***Figure 1*** *Global total anthropogenic emissions of SO₂, BC, OC, NOx, NH₃, and NMVOC in the CEDS21,*
*ECLv6, CEDS17 inventories, for the period 1990 to the most recent inventory year (2019, 2016 and*
*2014, respectively). Dotted lines show emissions from the SSP2-4.5 scenario, linearly interpolated from*
*2015 to 2019.*



***Figure 2*** *Absolute difference in regional mean burden of the key anthropogenic aerosol species between*
*simulations with CEDS21 and CEDS (upper bar) and ECLv6 and CEDS (lower bar). Regions are the*
*same as in Lund et al. (2019): EAS = East Asia, SAS = South Asia, SAF = Sub-Saharan Africa, NAM =*
*North America, SAM = South America, NAF = North Africa and the Middle East, EUR = Europe, SEA*
*= South East Asia, RBU = Russia.*

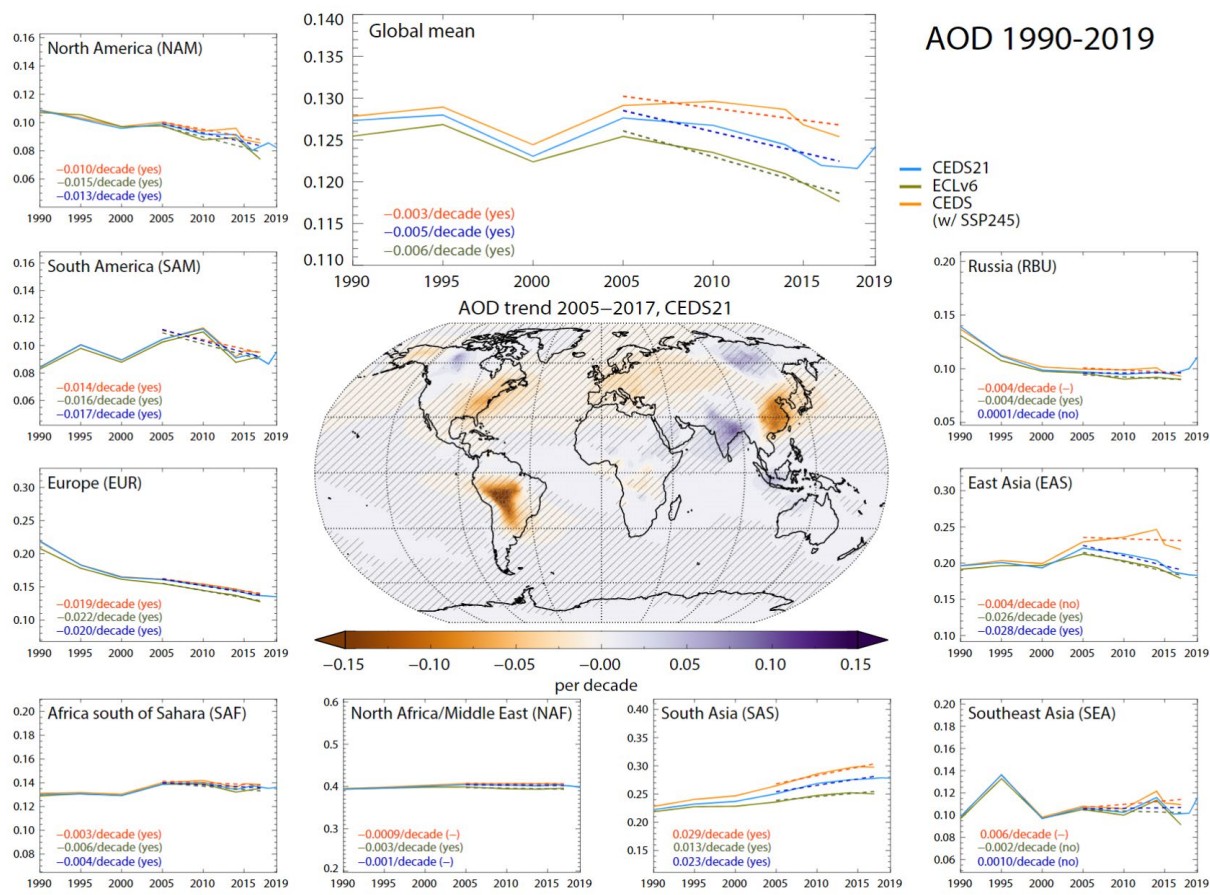



*Figure 3: Global and regional mean total AOD simulated with emissions from the CEDS21, ECLv6 and*
*CEDS inventories. In the case of CEDS, the timeseries is extended from 2014 to 2017 using SSP2-4.5*
*emissions. Dashed lines show the linear 2005-2017 trend, defined as statistically significant from no*
*trend when the linear Pearsons correlation coefficient is significant at the 0.05 level. To reduce any*
*influence of individual, outlier years on the trends, we calculate a set of trends removing one-and-one*
*year from the sample and show the average. Significance is given in the parenthesis. If a dash is given,*
*individual trends from the sample differed from each other in terms of significance.*










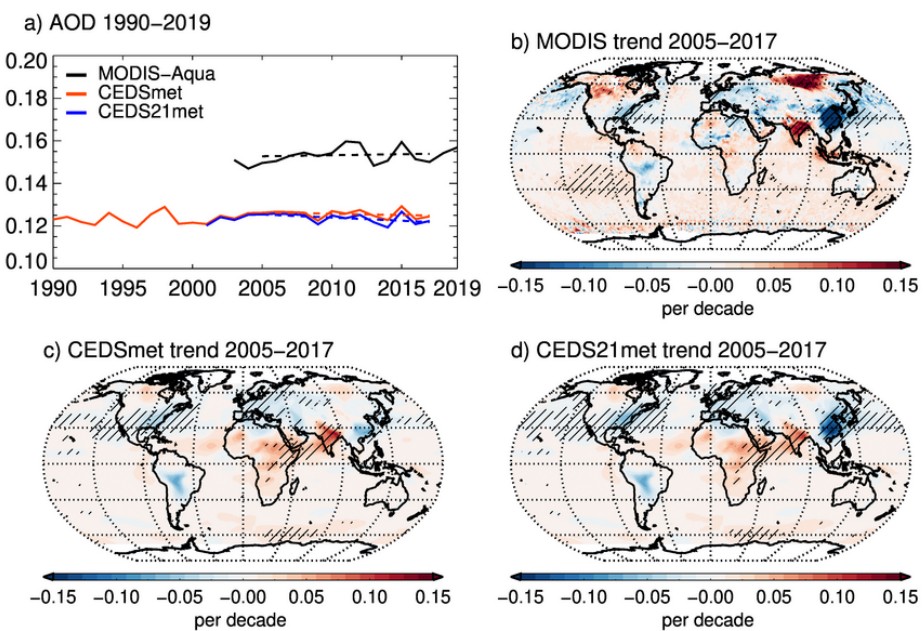



*Figure 4: a) Global, annual mean AOD from MODIS-Aqua and the OsloCTM3 over the 1990-2019 period. Note that data north and south of 70° is excluded here due to the limited MODIS-Aqua coverage. Dashed lines show linear trend from 2005 to 2017. b-d) Spatially resolved linear trends in observed and simulated AOD. Hatching indicates where the linear trend is significantly different from zero at the 0.05 level.*






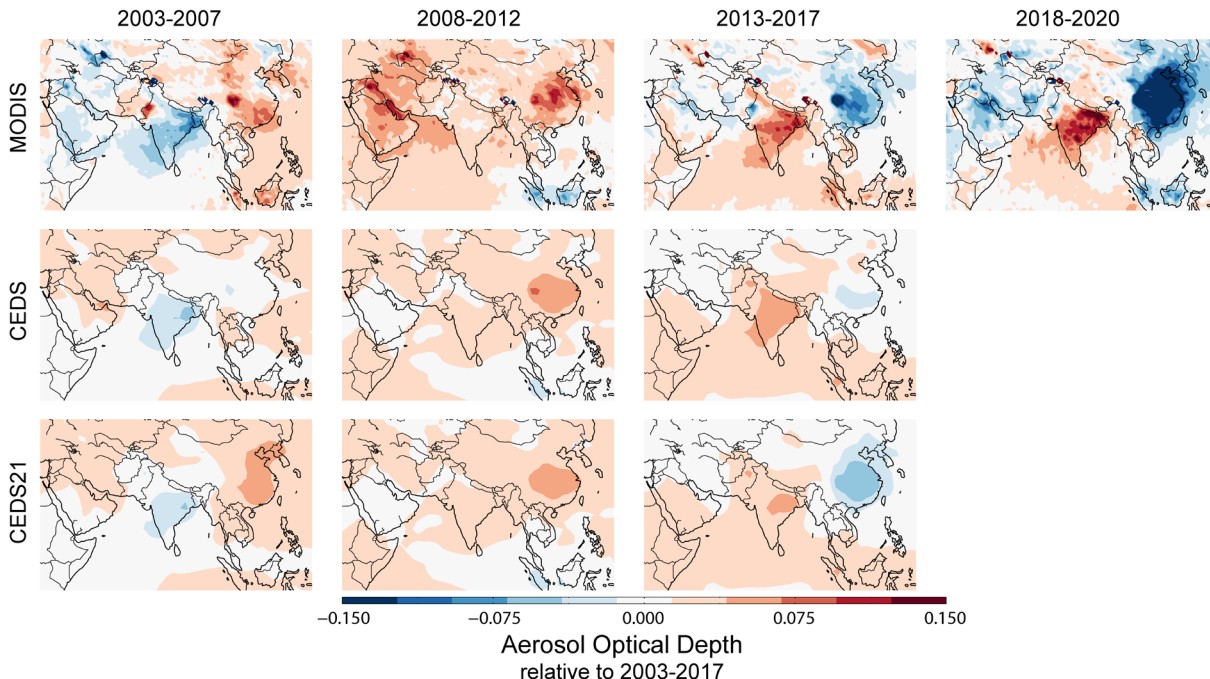


*Figure 5: Evolution of AOD over South and East Asia, and the Middle East, over the period 2003-2020. All panels show five-year average deviations from the period 2003-2017, except the rightmost MODIS-Aqua panel which show the three-year average deviation (same baseline). The top row shows retrievals from MODIS Aqua; the two bottom rows show model calculations with OsloCTM3 based on the CEDS and CEDS21 emission inventories.*

971

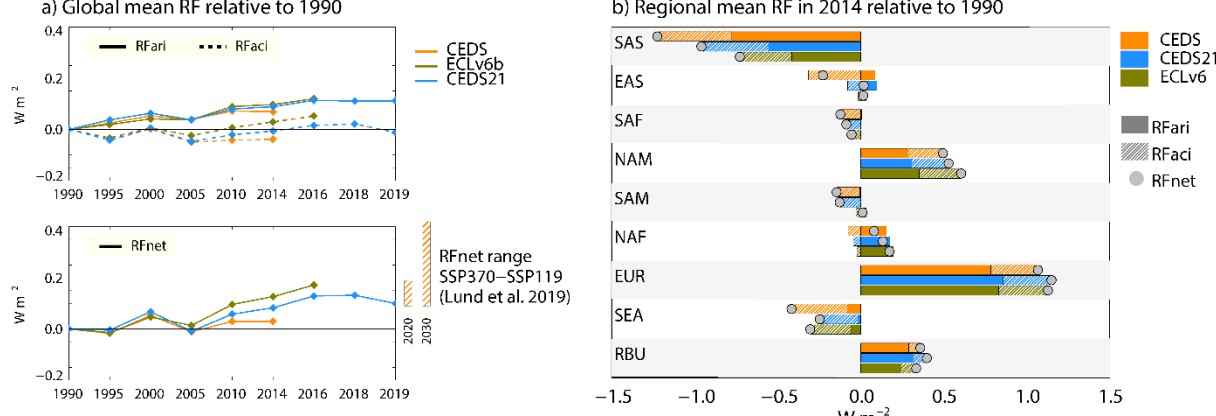

972

973

*Figure 6: a) Global mean RFari and RFaci (top) and RFnet (RFari+RFaci) (bottom) relative to 1990 from simulations using the CEDS, CEDS21, and ECLv6 emission inventories. The vertical bars to the right show the range in RFnet in 2020 and 2030 (relative to 1990) estimated with the SSP1-1.9 and SSP3-7.0 emissions (adapted from Lund et al. (2019)). b) Regional mean RFnet, RFari, and RFaci in 2014 relative to 1990 in simulations with CEDS, CEDS21, and ECLv6 inventories.*

979