# Peer review of "Implications of differences between recent anthropogenic aerosol emission inventories"

_Atmospheric Chemistry and Physics, 2022_

## Referee Comment (RC2)

**Reviewer Comments on Lund et al., 2022 - Differences between recent emission inventories strongly affect anthropogenic aerosol evolution from 1990 to 2019**

This study presents a comparison of the simulation of aerosols and aerosol radiative forcing over recent decades using a chemical transport model with three different anthropogenic emission inventories. A comparison is made of the simulation using the CEDS emission inventory (used in CMIP6), with two more recently created inventories: ECLIPSE v6b and CEDS21, a recent updated version of CEDS. A comparison is also made to MODIS AOD observations and surface AERONET AOD observations. The results show that simulations using CEDS underestimated magnitude of the total column aerosol burden and aerosol optical depth, particularly over east Asia, compared to the other inventories. In addition, the recent declining in aerosols and aerosol radiative forcing since 1990 is underestimated in these simulations with CEDS. Overall, the study highlights that using the updated anthropogenic emission inventories can better represent recent changes in aerosols, although a model bias in the absolute values of AOD still exists, and their impact on climate, via changes to the radiative balance.

I found this paper well written, with clear graphics and tables. However, it did seem to lack some additional details and more information on the causes and implications of these changes. I have provided some comments below which look to set out these points and where the manuscript could be improved to help the understanding of the topic further.

**Major Comments**

1. A reoccurring theme within this paper was the influence of biomass burning emissions (and also other natural emissions e.g., sea salt) on particular years when comparing trends between simulations and observations e.g., lines 238-239 for biomass burning and lines 323-234. Since these factors appear to be leading to reoccurring issues in some of the comparisons then it would make the manuscript better if more consideration could be given to dealing with some of these issues and perhaps removing the influence of particular years with high biomass burning emissions e.g. 2019. Or perhaps could a comparative simulation be performed when the biomass burning emissions are set to a fixed climatology to eliminate their influence? A further question arises in that are the linear trends presented in this analysis impacted by the choice of the start year and end year, and could there be a better way of removing the influence on trends of a particular year?

2. Uncertainty is mentioned in the manuscript in terms of observations and model simulations and providing context for the differences between them (e.g. interannual variability of MODIS observations – Fig 3b). I think it would make the figures better if uncertainty could be represented on the figures (e.g. Figure 5), and could better put into context how big or important any differences are. In addition, could some more background on aerosol radiative forcing uncertainty over recent decades (e.g., Regayre et al., 2014) be put into the introduction section to better frame this study.

3. The dipole pattern of radiative forcing differences between South Asia and East Asia is mentioned a number of times in the manuscript but I think the importance of this pattern is not really mentioned. Therefore, I think some text in the manuscript could improve this by identifying what does this pattern mean, why is it important and what are the implications for climate.

4.  Would it be best to put the AOD comparison with observations (Section 3.4) at the start of the results section or after the current section 3.2 to show the differences in the model performance over time before then going on to discuss how the different inventories impacts radiative forcing? Also it took me a while to figure out which simulations where used in each section so could this be made clearer throughout the manuscript. Line 305-206 states that the transient simulations are used here but is this the only place they are used?

5.  The change of increasing nitrate aerosols in the new inventories in most regions is quite interesting. The text in the manuscript gives a suggestion of why this bit the case but it would be good to have more details on why this occurs and in particularly why does nitrate aerosol decrease over south Asia? This could be of particular interest given that a lot of the CMIP6 models do not include NO3.

6.  The manuscript highlights the dominance of aerosol radiation interactions in the radiative forcing calculations. But is this a result of the method that you have use to calculate the radiative effects? If your simulations are nudged and you are using offline radiation calculations, then you wouldn't you expect the calculated aerosol-cloud interactions to be smaller? More comments on the split between the aerosol-radiation and aerosol-cloud interactions would be good in the manuscript. In particular, is the radiative forcing response over South America in Figure S3 due to cloud interactions?

**Minor Comments**

Line 38 – remove "and" and replace with comma

Line 40 – replace "dominating" with 'dominant'

Line 41 – state year that this plan was put in place and emissions began to decline. Also did the action plan specifically target these gases or was it just a target to reduce PM2.5 concentrations?

Line 45 – Is there not also strong growth in other air pollutant emissions over South Asia?

Line 49-50 – can you give examples of studies showing how aerosol impact regional climate?

Line 58-59 – Are BC and OC emissions lower everywhere in this updated version of CEDS?

Line 60 – Have the reductions in NOx emissions over China (stated on line 42) also been addressed in the revised CEDS emissions?

Line 96 – Perhaps it might be useful to put a small comment here on the evaluation of the model's present-day performance in simulating aerosols with the original CEDS inventory so the reader is aware of existing biases in the model for any particularly aerosol components.

Line 104 – Again perhaps a brief mention of the method used in Quaas et al., (2006) would be useful here for the reader to understand how the change in aerosol in OsloCTM3 is linked aerosol cloud interactions (i.e., an assumed relationship between AOD and CDNC).

Line 132 – Can you state the SSPs used and up to what time period? Also why 3 SSPs are used and are there large differences in anthropogenic aerosol emissions over the time periods of interest?

Line 137-138 – Can you state the natural emissions are kept fixed?

Line 147 – Tables S1 seems to show aerosol burdens and not a list of experiments as stated

Line 160-162 – Is the fact that some regional emissions are quite different pre-2000 important, especially given that your time series simulations with the new inventory only start in 2001 (line 146)?

Line 177-179 – What are the differences between these two sets of numbers? Is it the inclusion of biomass burning emissions?

Line 179 – This is currently Table S1 in the supplement

Line 181 – How small are the reductions in SOA?

Line 184 – Are there differences between anthropogenic VOCs in the inventories which could affect SOA formation? If so the differences in VOCs have not been discussed.

Line 189 – 190 – Is it useful to average the differences in burdens across regions? Doesn't that just take away from the importance of the regional differences? Perhaps best just to identify the largest differences in burden for each aerosol component or give a range of the differences across regions.

Line 199 – There has not been much of a discussion on any differences in NH3 emissions between inventories which could be important for this change in nitrate aerosols.

Line 200-201 – This is true what you have said but I am not sure it explains the increase in nitrate aerosols across nearly all the regions shown on Figure 2. For instance SO2 emissions have been reduced over South Asia in CEDS21 as well but this has not increased nitrate aerosols. This could do with more explanation.

Line 204-206 – Can you expand further on this point as to why it is important. Are you saying that the uncertainty in the inventories are as large as the trend in emissions?

Line 222 – so this means that they are significant differences?

Line 232 – replace "up" with 'higher'

Line 232-233 – If 2019 is a significantly higher biomass burning years does this then create problems when using it as the end year for the calculation of linear trends?

Line 246-247 – I think need to state some evidence to support the statement that "real world emissions have tracked below ssp245" over Asia.

Line 248-249 – Seems a shame to not carry on the discussion of regional trends now.

Line 259 – Is there a reason for the strong negative forcing over South America between 2014 and 1990? I notice AOD has increased but is this related to anthropogenic emission changes?

Line 269-271 – I think you need to refer to Fig S3 in this sentence as well. These are the plots that highlight the shift in emissions with the shift in radiative forcing.

Line 277 – replace "until" with 'by'

Line 297 – Does also this also depend on the speed of the reductions that are occurring in particular regions?

Line 326 – from Figure 5 can you really suggest that there are positive or negative trends in this data? I think you would have to suggest they are very small.

Line 328-329 – what is the trend in the ground based observations and how does it compare?

Line 329-331 – Has there been an observable increase in seas salt aerosol across the oceans across this time period? Since the focus of this paper is on anthropogenic emission inventories is it better to discount the influence of the ocean and take a land-only global mean trend?

Line 332 – is there a reason for the sudden increase in interannual variability in the MODIS observations?

Line 336-341 – There is a strong focus on sea spray changes in studies. Can we be sure that the positive MODIS trend is due to natural aerosols with a contribution from sea salt (and as stated in the conclusions)?

Line 358-359 – Can natural emissions and long-range transport really contribute that much to the large changes seen over Asia between 2018-2020 in MODIS? Is 3 years two short a window for comparison then?

Line 384 – Instead of the global evaluation plots could you also show the AOD evaluation across regions as it would highlight different areas of change and where the change in emissions has caused differences. Also could errors is aerosol process representation also be contributing to these underestimations? I am not sure it is all due to emissions.

Line 416 – "easter" should be 'Eastern'

Figure 2 – Need to state what year these differences are for.

Figure 3 – Can you make the linear trend lines a bit more distinct? Also panels in c) are currently labelled as b)

Figure 4 – can you make the colour bars wider so it easier to see the different shades

---

## Author Comment (AC1)

Response to Anonymous Referee #1 of Lund et al. 2022

We thank the referee for the review of our manuscript and suggestions for improvements. Responses to individual comments are given below.

More substantial changes in response to both referees include:

- Table S1: simplified to show the relative differences discussed in the main text instead of absolute numbers that were not really used.
- Updated Figures 1 and SI1 to include NH3 and VOCs.
- Updated Figure 3 and 4 to include more regionally explicit information.
- Trend calculations refined to reduce influence of individual years.
- Moved radiative forcing section to after evaluation of AOD against observations.
- Larger modifications to the description of simulated AOD trends and RF following comment from referee #1 on inclusion of regional information.

**General comments**

This paper reports the effect of three different inventories on aerosol optical depth and radiative forcing simulated with one model. Two of the inventories are different versions of CEDS. Both magnitude and trend of aerosol optical depth are compared with MODIS.

I find the contribution of this paper to scientific understanding to be rather modest. The use of models to understand how anthropogenic emissions affect the atmosphere's aerosol content and climate is of course a worthwhile pursuit. But an interesting work would include careful diagnosis of the causes of difference, or their implications for radiative forcing and climate response in different regions. This work doesn't provide that.

We thank the reviewer for the feedback and the suggestion. However, we do not agree that a diagnosis of the underlying causes between inventories is the only relevant question to be answered. It remains a fact that the inventories, as they are provided, are used in many applications, as documented in our introduction. Given this situation, a quantification of the uncertainty this introduces to current projections is of high importance for the community. We have now made this framing, and key question, clearer in both the title and introduction of our manuscript.

We do, however, agree with the reviewers that more regional detail beyond the full 2D maps can be of interest for the reader. As we're sure the reviewer is aware, we are not able to diagnose regional climate responses beyond radiative forcing within the modeling framework used in our study. Such quantifications typically require more resource demanding simulations with coupled models. We therefore see our study as an important first step that, given the differences we find, can motivate and support spending the time on further work.

To better highlight the regionality, we have produced two new figures where we quantify and show the regional mean AOD and trend, and the regional mean RF (see end of this document). Here we use a set of regions broadly covering the globe and largely in agreement with the study by Lund et al. (2019;2020), as well as the coarser set of IPCC AR6 region definitions. The text has been modified accordingly, as well as in response to the comments below (see respective responses). We think this

has significantly improved the manuscript and provides information of broader relevance to the more of the community.

Authors argue that the later version of CEDS was not included in AR6 and therefore it is worthwhile to analyze implications of the new inventory. That may be true, but the paper simply reports averages and shows spatial distributions. It doesn't provide much understanding of how or why the new inventory is different or whether it is more or less suitable to represent anthropogenic influence.

As described above (and below), we have added more detail about the regional changes and attempt to link them better to the underlying emission changes (and reasons, where such information is available). We also connect these regions better to the description of observed AOD trends, although the latter is of course influenced by more than anthropogenic emission and these factors can sometimes obscure the detection of anthropogenic emission-driven trends. Moreover, our existing comparison of the simulated AOD with MODIS over the Asian region where key emission biases are known to exist (changing estimates of Asian aerosol emissions and evolving inventories was a challenge for the assessment of climate change throughout the sixth assessment report of the IPCC) does confirm that our model is better able to represent the trends (and hence anthropogenic influence) with the updated emissions.

To the extent it's possible to derive from the limited documentation, we give a summary of the underlying updates to the inventories, with associated references. See also further responses to the comments regarding inventories below.

**Specific comments**

Emission inventories do not affect actual aerosol influence (as is suggested in the title), rather simulated influence. The purpose of a model is to attempt to reflect the real evolution. Certainly if one changes any flux (emissions) or any input then it changes the simulation, but this shouldn't be a surprise if one's model is working properly. Perhaps some understanding could be gained by exploring whether the model outputs (AOD, RFari, RFaci) scale with the inventory changes. This sort of analysis is hinted at, e.g. in lines 207-208, but for a helpful contribution to the community, much more analysis would be presented.

We see the point about the wording not being accurately representative for what we do here. The title has been modified to make it more precise in wording, now reading:

*"Implications of differences between recent anthropogenic aerosol emission inventories on diagnosed AOD and radiative forcing from 1990 to 2019"*

We have also made updates to introduction to better reflect this same point, as well as to clarify our aims and motivation.

Yes, changing any one flux or input changes the simulation. This is true for any modeling experiment. However; one needs to do the actual simulation to understand exactly how the change manifests. It is not given that the model will perform better with changed data, as there may be compensating issues affecting biases compared to observations, or competing effects when there are concurrent differences across emitted species. Furthermore, while some atmospheric species are more linearly connected to emissions, others are affected by complex atmospheric chemistry. The effect on atmospheric composition of equal changes in emissions can also be regionally dependent, as is the resulting radiative forcing.

We believe that the inclusion of more regionally explicit information as suggested by the reviewer, helps to highlight the importance and implications of our work. Throughout we have also made modifications to the text to better link the results with the input data, e.g. linking regional burden and AOD of individual species better with the underlying emission changes to see if they scale or not.

Unfortunately, we do not have the RF per component available, so linking forcing to inventory changes is not straightforward. We have, however, looked into the RF per dAOD, a measure sometimes used for the intercomparison across different models, for both global and regional means. Across different regions, this measure of course varies substantially due to the different aerosol composition, but also due to background climate and underlying surface characteristics. Within a given region, we also, in some cases, find marked differences in the normalized RF between the experiments using different emission inventories. For instance, the normalized RF averaged over Russia is consistently lower (by up to 20%) with CEDS21 and ECLv6 emissions than with CEDS, while for other regions both higher and lower normalized RF values are found depending on whether RFari and RFaci is considered. This likely reflects the complexity of the response when multiple emitted species are changed at the same time and sometimes with different signs. In some cases, these normalized numbers are also difficult to interpret given their sensitivity to the denominator. In particular where the AOD change is very small, the numbers can become very high. However, the diversity across region and inventory adds to the importance of conducting detailed simulations and not try to derive the response directly from the inventory change.

Emissions are a component of the physical system that are affected by processes. This paper compares different compilations of or assumptions about those processes. But ascribing these differences to the compilation label itself, eg 'CEDS', 'ECLIPSE' is overly simplistic. What assumptions have the inventory developers made that cause these differences?

The objective of this paper is to document results from simulations using some of the central recent inventories provided to the modeling community. The names of inventories are used to label the experiments in the analysis, and we describe and quantify the differences between these experiments using different input data. To try to clarify, we added a table describing the experiments and their names.

The modeling community is dependent on the provision of up-to-date emission inventories. However, their development is comprehensive and complex work, and the documentation can be limited. Hence, it is often difficult for non-experts to know the numerous underlying assumptions and data or understand changes. We would maintain that not being able to attribute differences in emission trends or magnitudes uniquely to specific changes in assumptions or bottom-up statistical data does not take away from the importance of quantifying the impact of these differences on the diagnosed quantities critical for assessing the air quality and climate implications of anthropogenic aerosol. In fact, rather the opposite, as it is not possible to assess the consequent changes in aerosol composition and distribution (nor RF) directly from the inventory differences, e.g. via scaling by emission changes. This requires the type of detailed modeling we have performed.

A comprehensive documentation of the assumptions that inventory developers have made or which data they have updated is beyond the scope of this study. We certainly think that such a study would be very helpful for the modeling community but would argue that this is a task for the inventory developers. In fact, a recent opinion piece by Smith et al. highlights the importance of development of consistent and transparent inventories for pollutants, which is not the case today.

However, some first order information about the link between emission changes and underlying updates can be derived from the limited literature that is available. We have expanded the paragraph

describing the emission inventories. Where possible and most relevant, we also try to link the observed difference trends more clearly with underlying emission changes. By providing more regional information, in response to the comment above, this is better facilitated. Finally, we try to better highlight our aims and motivation in the introduction section, and point to why such simulations are needed (i.e. that the information cannot be simply obtain from scaling by emissions) in the results description.

Aerosol trends are discussed in lines 234-249 and 342-364. This sort of analysis could aid in identifying, explaining and quantifying differences in the input (emissions) and response (AOD, RFari). But the analysis presented here is rather broad ('weaker in magnitude', 'consistent with…') What are the implications for RFari and RFaci, since a global average is given for these measures relative to 1990?

We have quantified regional trends in AOD as well as regional mean RFs in the revised manuscript, where possible describing the connection with the underlying emission trend better.

The presentation also ascribes some masking of trends to interannual variability, especially among natural (sea-salt) or biomass burning emissions. This is a well-known issue in comparing model results with observations. It would seem that model evaluators should have some set of best practices about how to account for this effect after many years of such studies, such as using running means. Otherwise, the persistent inability to draw conclusions will render all such studies only marginally useful. Do authors have thoughts on this?

We certainty agree that model comparisons with observations of various spatiotemporal resolutions and type is not straightforward, which is also well known in the community. Many approaches exist (we for instance use a linear least square fitting and 10-year boxcar average), but if there are opportunities to unite the community better around a common, optimal approach, that seems like an effort that is worthwhile to pursue in further work.

In the revised manuscript, in response also to comments by referee #2, we have changed the calculations of trends to minimize the influence of single years or start-end yeas (mainly due to the influence of biomass burning, since our dust and sea salt fluxes are kept constant in the fix met runs). We now calculate trends with one and one year removed and show the average of this set of coefficients. We note that the difference is small and the change in method does not affect our overall findings. We have also made some modifications to the paragraph on sea salt following referee #2.

It may be worthwhile to define how well one needs to know the forcing since emissions and other inputs are always uncertain. Then one would have more confidence in stating a 'strong effect' as is done in the title.

This is an interesting question and one we don't have a clear answer to. However, we agree that the use of "strong" in the title can be perceived as subjective without a definition such as suggested by the referee. In light of this and the comment above, we have modified the title.

The paper acknowledges analysis by other researchers on the same topic, e.g. Lund et al 2018, Mortier et al 2020, Quaas et al 2022. However, other than broadly comparing findings, this work doesn't indicate what new insights it has offered – what has been done here that wasn't done before, and if another future paper is done with similar approach, what questions should it attempt to answer? There seems to be a limited review of prior work, especially considering that Asian emission inventories have been evaluated against observations, and those emissions are also stated to play an important role in this work.

We have modified the introduction section to clarify our motivation and aims. Upon submission this was, to our knowledge, a first paper comparing these three inventories which makes it difficult to compare directly with other literature. Furthermore, our objective/scope is not to provide a review of emission inventories or aerosol trends, but rather to evaluate our modeling tool, assess whether the availability of new inventories improves the model performance or highlights other issues, and understand to what extent the known biases in emissions have affected assessed aerosol-induced climate effects (or may even continue to do so given that studies are still published using the older generation emissions data). We have tried to make this clearer in the introduction and conclusion. Furthermore, we have added more references to aerosol uncertainty and climate effects and work on Asian emissions.

**Technical comment**

I found the paper well written and I did not note technical corrections to make it clearer or more accurate.

Thank you.

Draft new figures 3 and 4 (larger size, better resolution in revised manuscript):

---

## Author Comment (AC2)

**Response to Anonymous Referee #2 of Lund et al. 2022**

This study presents a comparison of the simulation of aerosols and aerosol radiative forcing over recent decades using a chemical transport model with three different anthropogenic emission inventories. A comparison is made of the simulation using the CEDS emission inventory (used in CMIP6), with two more recently created inventories: ECLIPSE v6b and CEDS21, a recent updated version of CEDS. A comparison is also made to MODIS AOD observations and surface AERONET AOD observations. The results show that simulations using CEDS underestimated magnitude of the total column aerosol burden and aerosol optical depth, particularly over east Asia, compared to the other inventories. In addition, the recent declining in aerosols and aerosol radiative forcing since 1990 is underestimated in these simulations with CEDS. Overall, the study highlights that using the updated anthropogenic emission inventories can better represent recent changes in aerosols, although a model bias in the absolute values of AOD still exists, and their impact on climate, via changes to the radiative balance. I found this paper well written, with clear graphics and tables. However, it did seem to lack some additional details and more information on the causes and implications of these changes. I have provided some comments below which look to set out these points and where the manuscript could be improved to help the understanding of the topic further.

We thank the reviewer for the thorough and positive review, and for the constructive suggestions and comments. We believe that in addressing them, the manuscript has been significantly improved. Please see detailed responses and associated changes below.

More substantial changes in response to both referees include:

- Table S1: simplified to show the relative differences discussed in the main text instead of absolute numbers that were not really used.
- Updated Figures 1 and SI1 to include NH3 and VOCs.
- Update Figure 3 and 4 to include more regionally explicit information (following comments by referee #1). Trend calculations refined to reduce influence of individual years.
- Moved radiative forcing section to after evaluation of AOD against observations.
- Larger modifications to the description of simulated AOD trends and RF following comment from referee #1 on inclusion of regional information.

Major Comments

1. A reoccurring theme within this paper was the influence of biomass burning emissions (and also other natural emissions e.g., sea salt) on particular years when comparing trends between simulations and observations e.g., lines 238-239 for biomass burning and lines 323- 234. Since these factors appear to be leading to reoccurring issues in some of the comparisons then it would make the manuscript better if more consideration could be given to dealing with some of these issues and perhaps removing the influence of particular years with high biomass burning emissions e.g. 2019. Or perhaps could a comparative simulation be performed when the biomass burning emissions are set to a fixed climatology to eliminate their influence? A further question arises in that are the linear trends presented in this analysis impacted by the choice of the start year and end year, and could there be a better way of removing the influence on trends of a particular year?

Removing the influence of biomass burning by setting it to a fixed climatology is a good suggestion, however, a set of new simulations is beyond the resources available for this study. Better disentangling

the influence from biomass burning, including how much can be considered anthropogenic and how much wildfire contribute to too high air pollution levels regionally, would certainly be an interesting follow-up study. Other natural aerosols, mainly sea salt and dust, remain unchanging in the simulations with fixed meteorology.

While additional simulations are not possible here, we have, however, adopted a different approach to the trend calculations. To minimize the influence of start and end year or individual high/low AOD years, we have in the revision calculated the trend with one and one year removed from the sample, i.e. producing 13 estimates of trend over 12 years in the 2003-2017 period. The updated figures show the average of these. Overall, we find only small differences between the 13 trends and only in a couple of regions does the conclusions about significant linear trend vary between the 13 different estimates. The methods section is updated accordingly.

2. Uncertainty is mentioned in the manuscript in terms of observations and model simulations and providing context for the differences between them (e.g. interannual variability of MODIS observations – Fig 3b). I think it would make the figures better if uncertainty could be represented on the figures (e.g. Figure 5), and could better put into context how big or important any differences are. In addition, could some more background on aerosol radiative forcing uncertainty over recent decades (e.g., Regayre et al., 2014) be put into the introduction section to better frame this study.

Since we have only one model and single time series with the individual inventories, a comprehensive assessment and inclusion of uncertainties across the different quantifies is of course challenging. Moreover, several uncertainties, such as relating to atmospheric processing of aerosols or in the optical properties of the aerosol, are also systemic and would apply equally to simulations using all three scenarios. For others, and especially uncertainty in the emissions themselves, there is very limited consistent information available as this is typically not provided as part of the inventory.

The comparison of spread in modeled AOD for a single year with interannual variability in MODIS is meant to provide context for the magnitude but is not discussed in terms of it being an uncertainty. Moreover, the role of interannual variability is however quantitively included in the assessment of significance of the linear trends and is already shown in Fig. 5b-d through the hatching. Further quantitative uncertainty estimates are included through the RMSE and NMB calculated for the comparison with AERONET stations.

There is of course uncertainty in the remote sensing data, which is not visible in the current figures. To our knowledge, the MODIS product does not include an uncertainty range. In the recent study, Vogel et al. 2022 found a 13% spread in AOD between different satellite products, adding to any retrieval uncertainty – on a global scale. We don't have corresponding information for the regional level, however, to make the discussion more quantitative, we do add the numbers from Vogel rather than just a general statement, and related the AOD differences better to this spread.

Finally, in the discussion of RF, we have also added a reference to uncertainty over recent decades (as suggested, but in a different section).

3. The dipole pattern of radiative forcing differences between South Asia and East Asia is mentioned a number of times in the manuscript but I think the importance of this pattern is not really mentioned. Therefore, I think some text in the manuscript could improve this by identifying what does this pattern mean, why is it important and what are the implications for climate.

We have expanded the sentence in the introduction with references to the role of aerosols. The key point is also that the implications are for climate is not fully known and improving that understanding require robust emission estimates.

*"Such rapid aerosol changes are likely to affect the climate of the region, as aerosols have been shown to have a notable influence on regional temperature and precipitation, including extremes (e.g. Bollasina et al., 2011; Hegerl et al., 2019; Marvel et al., 2020; Samset et al., 2018; Sillmann et al., 2013), with different responses to scattering and absorbing aerosols. However, the exact nature and magnitude of the climate implications need to better quantified (Persad et al., 2022)."*

4. Would it be best to put the AOD comparison with observations (Section 3.4) at the start of the results section or after the current section 3.2 to show the differences in the model performance over time before then going on to discuss how the different inventories impacts radiative forcing? Also it took me a while to figure out which simulations where used in each section so could this be made clearer throughout the manuscript. Line 305-206 states that the transient simulations are used here but is this the only place they are used?

We did indeed consider having the AOD comparison with observations before the RF in earlier version of the manuscript. In response to the referee comment, we now adopt this structure and move the description of radiative forcing to the end.

Regarding different simulations: To try to make it easier to follow the experiments used, we have added a table the different time series in the methodology section. We have also added an associated paragraph:

*"Five different time series of simulated aerosol distributions covering the 1990-2019 period are included in this analysis, using three different emission inventories and either fixed meteorology or meteorology corresponding to the emission year. The fixed meteorology runs forms the basis for investigating differences in simulated anthropogenic aerosol and corresponding RF, while the latter is used in the comparison with observed AOD. Table 1 provides a summary of the experiments."*

5. The change of increasing nitrate aerosols in the new inventories in most regions is quite interesting. The text in the manuscript gives a suggestion of why this bit the case but it would be good to have more details on why this occurs and in particularly why does nitrate aerosol decrease over south Asia? This could be of particular interest given that a lot of the CMIP6 models do not include NO3.

We have expanded the description of the change in nitrate aerosol and extended Figure 1 and S1 to also include NH3 emissions. The main paragraph now reads:

*"The only species that is globally more abundant in simulations with the two new inventories, is nitrate. There is considerable regional heterogeneity, where the burden is lower compared to the CEDS experiments in South Asia and on the US east coast but higher in the US Midwest, parts of Africa and South America, and, especially, over East Asia (Fig.2, Fig.S2). While absolute differences are small in many regions compared to other species, the net effect is nevertheless a 15 and 24% higher global mean nitrate burden with CEDS21 and ECLv6, respectively, compared to using CEDS emissions. Changes in the atmospheric nitrate distribution results from a complex interplay between differences in emissions of NOx, NH3, and SO2. Studies have also shown that nitrate formation can be influenced by background concentrations of VOCs (e.g. Womack et al., 2019) We find the largest absolute difference in nitrate in East and South Asia, however, of opposite sign. In East Asia, emissions of SO2 and NOx are both lower in ECLv6 and CEDS21 than in CEDS, whereas NH3 emissions are higher (Fig.1, Fig.S1). This results in lower chemical competition for available sulfate and, in turn, enhanced formation of nitrate*

*aerosol. In South Asia, SO2, NOx, and NH3 are all lower in the two new inventories than in CEDS, as is the nitrate burden. Differences in concentrations of VOCs in the simulations with different inventories is a further complicating factor. Studies have suggested that nitrate formation can be more sensitive to changes in VOCs than NOx, however, this is highly site specific (Yang et al., 2022). Further delineating the role of individual factors on nitrate differences would require simulations beyond what is available for the current study. The potential for an increasing relative role of nitrate for air pollution and climate in a world with concurrent declines in SO2 and NOx emissions but little in NH3 has also been discussed in previous studies (e.g. Bauer et al., 2007; Bellouin et al., 2011; Zhai et al., 2021). However, while more studies have focused on local air pollution impacts of nitrate, and associated mitigation strategies, nitrate is still missing from many global climate models. Moreover, when included the model diversity in simulated distributions is large (Bian et al., 2017). Our results suggests that uncertainties in emissions and use of inventory can contribute to spread in simulated nitrate aerosols and confound the comparison of conclusions across modeling studies. Moreover, the complexity of the nitrate response demonstrates that the impact of inventory differences on simulated aerosols cannot be understood from scaling with the changes in individual emissions but require explicit modeling."*

6. The manuscript highlights the dominance of aerosol radiation interactions in the radiative forcing calculations. But is this a result of the method that you have use to calculate the radiative effects? If your simulations are nudged and you are using offline radiation calculations, then you wouldn't you expect the calculated aerosol-cloud interactions to be smaller? More comments on the split between the aerosol-radiation and aerosol-cloud interactions would be good in the manuscript. In particular, is the radiative forcing response over South America in Figure S3 due to cloud interactions?

We do consistently show also both RFari and RFaci, and the intent was not to really highlight any dominance of one or the other. However, it is a good point that our framework only allows us to include an estimate of the cloud albedo effect and may underestimate the full aerosol-cloud radiative effect. We have added a sentence to the section describing RF:

*"We note, however, that our framework only captures the cloud albedo effect and not radiative effects of any changes in cloud lifetime that may arise through the influence of aerosols."*

In response to referee #1's comment of lack of regional information, we have in the revised manuscript included a new RF figure which shows regional means. To better show the split in aerosol-cloud and aerosol-radiation, this figure is further split by RFari, RFaci, and net. This figure also confirms that the radiative forcing over is due to aerosol-cloud interactions, as also shown in Fig.S3. The text has been modified accordingly to describe this new figure, including, where relevant, a focus on differences between RFari and RFaci.

[Figure]

Minor Comments

Line 38 – remove "and" and replace with comma

Done

Line 40 – replace "dominating" with 'dominant'

Done

Line 41 – state year that this plan was put in place and emissions began to decline. Also did the action plan specifically target these gases or was it just a target to reduce PM2.5 concentrations?

Specification and references added. Sentence now reads:

*"However, since the adoption of the national action plans targeting particulate matter levels (i.e. Air Pollution Prevention and Control in 2013 (SCPRC, 2013) and Winning the Blue Sky Defense Battle in 2018 (SCPRC, 2018)), (…)"*

Line 45 – Is there not also strong growth in other air pollutant emissions over South Asia?

Yes, good point, thanks. Sentence slightly rephrased to:

*"A strong growth in emissions of SO2 and other pollutants has been seen in South Asia (Kurokawa & Ohara, 2020), resulting according to studies in India overtaking China as the dominant emitter of SO2 (Li et al., 2017)."*

Line 49-50 – can you give examples of studies showing how aerosol impact regional climate?

Added, along with a slight extension of the respective sentences:

*"Such rapid aerosol changes are likely to affect the region, as aerosols have been shown to have a notable influence on regional temperature and precipitation, including extremes (e.g. Bollasina et al., 2011; Hegerl et al., 2019; Marvel et al., 2020; Samset et al., 2018; Sillmann et al., 2013), with different response to scattering and absorbing aerosol. However, the exact nature and magnitude of the climate implications need to better quantified (Persad et al., 2022)."*

Line 58-59 – Are BC and OC emissions lower everywhere in this updated version of CEDS?

Within regions there are smaller areas of larger values in the updated version, however, in terms of both regional and global totals the updated BC and OC emissions are lower. We have modified the sentence to read:

*"Specifically, emissions of BC, OC and NOx are all substantially lower in the update, in global totals and particularly in Asia, and issues related to the decreasing trend in Chinese SO2 are largely addressed."*

(note however that the sentence has been moved to section describing emissions in response to other comments.)

Line 60 – Have the reductions in NOx emissions over China (stated on line 42) also been addressed in the revised CEDS emissions?

Yes, NOx emissions are also lower, both globally and in Asia, in the revised emissions. While this is further discussed in Sect. 2, we have also specified it here and modified the related sentence to:

*"Specifically, emissions of BC, OC and NOx are all substantially lower in the update, and issues related to the decreasing trend in Chinese SO2 are largely addressed."*

Line 96 – Perhaps it might be useful to put a small comment here on the evaluation of the model's present-day performance in simulating aerosols with the original CEDS inventory so the reader is aware of existing biases in the model for any particularly aerosol components.

As both the methods section and the introduction are quite extensive already and given that we cover this later in the model-observation comparison, we have chosen not to extend the section here.

Line 104 – Again perhaps a brief mention of the method used in Quaas et al., (2006) would be useful here for the reader to understand how the change in aerosol in OsloCTM3 is linked aerosol cloud interactions (i.e., an assumed relationship between AOD and CDNC).

Good suggestion. We have added:

*"Briefly, this approach is based on a statistical relationship between cloud droplet number concentrations and fine-mode AOD derived from satellite data from the MODerate Resolution Imaging Spectroradiometer (MODIS)."*

Line 132 – Can you state the SSPs used and up to what time period? Also why 3 SSPs are used and are there large differences in anthropogenic aerosol emissions over the time periods of interest?

Included, with paragraph now reading:

*"Results from the current study are compared with previously published results from simulations over 1990 to 2014 performed with the first release of the CEDS emissions (hereafter "CEDS") and three of the SSP scenarios (SSP1-1.9, SSP2-4.5, and SSP3-7.0) from 2015 to 2100 (here we use data for 2020 and 2030) (Lund et al., 2018; Lund et al., 2019). These three scenarios broadly span the range of aerosol and precursor emissions projected in the SSPs."*

Line 137-138 – Can you state the natural emissions are kept fixed?

This include dust and sea salt, as well as trace gases from marine sources, soils and vegetation and is now specified.

Line 147 – Tables S1 seems to show aerosol burdens and not a list of experiments as stated

We apologize for this mistake, at some point a second table fell out of the draft. We have added the table describing the experiments. However, in an attempt to make it easier to follow the different time series used, we have added this table to the main text.

Line 160-162 – Is the fact that some regional emissions are quite different pre-2000 important, especially given that your time series simulations with the new inventory only start in 2001 (line 146)?

There are indeed regional emission differences prior to year 2001. However, these time series are used for the MODIS comparison which only goes back to the early 2000's (see also response to major comment #4 where we have tried to improve the description on how the different data sets are used). To limit the computer resources, we therefore limit the updated time series to this period.

Line 177-179 – What are the differences between these two sets of numbers? Is it the inclusion of biomass burning emissions?

Yes, for BC and OC, the model has separate tracers for aerosols from biomass burning and biofuel and fossil fuel sources. Hence, numbers can be given for fossil fuel and biofuel sources only. We have tried to clarify in the text and Table S1 caption.

Line 179 – This is currently Table S1 in the supplement

Corrected, thanks for noticing this error.

Line 181 – How small are the reductions in SOA?

3-4% depending on inventory. Has been added to the text. Furthermore, we have changed Table S1 so that it actually summarizes these numbers rather than leaving it to the reader to calculate.

Line 184 – Are there differences between anthropogenic VOCs in the inventories which could affect SOA formation? If so the differences in VOCs have not been discussed.

We have for comprehensiveness added the total NMVOC and NH3 emissions from the different inventories to Figures 1 and S1. We have also modified this sentence to:

*"However, the SOA abundance is affected by the lower emissions of anthropogenic VOCs in both CEDS21 and ECLv6 than in CEDS (Fig. S1), as well as by lower amounts of POA, which serve as substrates for SOA formation, in simulations with the two new inventories."*

Line 189 – 190 – Is it useful to average the differences in burdens across regions? Doesn't that just take away from the importance of the regional differences? Perhaps best just to identify the largest differences in burden for each aerosol component or give a range of the differences across regions.

This sentence is meant to refer to regional averages not the averages over regions. To clarify, we have modified to "in these regions".

Line 199 – There has not been much of a discussion on any differences in NH3 emissions between inventories which could be important for this change in nitrate aerosols.

Yes, good point, thanks. We have added NH3 emissions to Fig. 1 and Fig.S1 for comprehensiveness and for the discussion on nitrate aerosol changes.

Line 200-201 – This is true what you have said but I am not sure it explains the increase in nitrate aerosols across nearly all the regions shown on Figure 2. For instance, SO2 emissions have been reduced over South Asia in CEDS21 as well but this has not increased nitrate aerosols. This could do with more explanation.

We agree that this section provides too little information on possible reasons. The section on nitrate has been expanded with more description – see response to major comment #5 above. While we do not have the simulations required to quantify the relative role of each factor, the discussion now includes NH3 and NOx emissions, as well as references to the potential role of NOx/VOC yields.

Line 204-206 – Can you expand further on this point as to why it is important. Are you saying that the uncertainty in the inventories are as large as the trend in emissions?

Yes, the point was to add some context to the spread in emissions/results. We have modified the sentences:

*"To place the spread between emission estimates into more context, we compare the differences in simulated aerosol burdens in 2014 to the difference in burdens over the 5-year period from 2014 to 2019 using CEDS21. Both globally and regionally, these differences are of the same order of magnitude. In other words, at least in this case, the spread between emissions is as large as the recent overall change in emissions."*

Line 222 – so this means that they are significant differences?

The interannual variability is added to provide some context to the magnitude of the changes, but we believe that significance cannot strictly be stated from this comparison. However, the comparison shows that the differences are larger in the main regions than the variation that can be expected from year to year and hence is important.

Line 232 – replace "up" with 'higher'

Done.

Line 232-233 – If 2019 is a significantly higher biomass burning years does this then create problems when using it as the end year for the calculation of linear trends?

It could. We have tested the impact of individual years by calculating trends with one and one year removed. For the long-term trend (full period, 1990-2019) in simulations with fix meteorology, we estimate a consistently significant negative trend. This also applies to most regional mean trends. However, in one case, for South East Asia, we do find that removing high biomass burning years (specifically 2019 and 2014), changes the trend from non-significant negative to significant negative. The same applies to the shorter 2005-2019 trend; global mean trend is not markedly affected, but the significance changes depending on year excluded for one region. So, in the combined case of weak regional trends and strong biomass influence, the trend can be affected. As already described above, we now show the average of the set of trends calculated with one and one year removed. This does not change any of our conclusions.

Line 246-247 – I think need to state some evidence to support the statement that "real world emissions have tracked below ssp245" over Asia.

Agreed, this was not formulated very precisely and of course applies under the assumptions that reported decline in emissions can be trusted and verified. Both bottom-up studies and remote sensing do suggest that is the case. Further support is given by the fact that our simulated aerosol distributions agree much better when we use the lower emission estimates. However, as the comparison with MODIS only comes later, we have removed the sentence here.

Line 248-249 – Seems a shame to not carry on the discussion of regional trends now.

The structure has now changed following the comment above, such that the flow from the simulated AOD to the comparison with observations is hopefully better.

Line 259 – Is there a reason for the strong negative forcing over South America between 2014 and 1990? I notice AOD has increased but is this related to anthropogenic emission changes?

A combination of factors. Over South America, we find an increase in the optical depth of BC, OA, and in particular nitrate over the period. We also find a weak increase in SOA, while sulfate is quite flat. In all three inventories anthropogenic emissions of NOx and NH3 increase, while for BC and OC, emissions increased weakly until 2014 and have in recent years leveled off or slightly declined. We also find an increase in biomass burning aerosol between 1990 and 2014 which contribute to the forcing. Following a comment from referee #1 we have looked into the RF per AOD change. For this region, we find stronger than average normalized RFaci, suggesting that it is quite sensitive. This and other regional information is better presented in the revised manuscript.

Line 269-271 – I think you need to refer to Fig S3 in this sentence as well. These are the plots that highlight the shift in emissions with the shift in radiative forcing.

Thank you for spotting this issue, reference added.

Line 277 – replace "until" with 'by'

Done.

Line 297 – Does also this also depend on the speed of the reductions that are occurring in particular regions?

Yes, the sentence was meant to capture speed and amount. We have rephrased to try and clarify:

*"Missing or incorrectly captured past emission trends can therefore markedly affect assessments of projected near-term aerosol-induced climate impacts, as they depend on a well constrained starting point."*

Line 326 – from Figure 5 can you really suggest that there are positive or negative trends in this data? I think you would have to suggest they are very small.

While the calculations do give a positive value, in contrast to the weak negative in the model data, we agree that this is a small number. We do state further down that the trend is not significant, however, for clarify, we have added "weak" and moved the significance up. The first sentences now read:

*"MODIS-Aqua data indicates a weak positive linear trend of 0.001 per decade in global mean AOD over the 2005-2017 period (0.004 per decade when extending the data to 2019). We do not, however, find this trend to be significant at the 0.05 level."*

Line 328-329 – what is the trend in the ground based observations and how does it compare?

This sentence contains an error and should have been removed from the manuscript before submission, we apologize for that. The trend it refers to is not from the observations (which do not provide the full global coverage) but from a climate model with full diagnostics used in the study. We have removed the sentence here.

Line 329-331 – Has there been an observable increase in seas salt aerosol across the oceans across this time period? Since the focus of this paper is on anthropogenic emission inventories is it better to discount the influence of the ocean and take a land-only global mean trend?

To our knowledge, no, a consistent, significant increase in sea salt aerosols has not been observed. In fact, since submission the corresponding author has become aware that MISR AOD retrievals do not show the same marine positive trend as MODIS. This is now also reflected in the manuscript – see response to comment below as well. Moreover, we see that the description of sea salt distracts from our focus from anthropogenic aerosols and have therefore reduced the length of the section, maintaining only a sentence on the uncertainty of sea salt trends.

Line 332 – is there a reason for the sudden increase in interannual variability in the MODIS observations?

As far as we can tell, there is no clear explanation for the larger variability in the latter half of the time series in the cited study.

Line 336-341 – There is a strong focus on sea spray changes in studies. Can we be sure that the positive MODIS trend is due to natural aerosols with a contribution from sea salt (and as stated in the conclusions)?

Oceanic regions with an observed positive trend in AOD are likely associated with an increase in sea spray aerosols, but this cannot of course not be confirmed from these data alone. Moreover, this is one contribution to the weak global trend, but there are also positive trends over India, Southeast Asia

and the boreal northern hemisphere regions. The point was to highlight the possible contributions, but we see that the text is not well enough written. We try to clarify and have modified to:

*"Regions of significant positive observed AOD trend include the Indian subcontinent, northern North America and Russia, and over parts of the ocean in the southern hemisphere (Fig. 5b). The two high latitude regions have recently seen unprecedented wildfire activity and GFED4 emissions show a positive trend over the 2005-2017 period here, suggesting that biomass burning aerosols play an important role for the observed AOD trend. Over the oceans, sea salt aerosols could be causing the increase. However, (Quaas et al., 2022) recently showed that this positive trend is not clear in MISR data."*

We have also modified the conclusion:

*"In particular, the model does not capture the strength of the positive AOD trend observed over high latitude North America and Russia, likely due to an increase in biomass burning aerosols."*

Line 358-359 – Can natural emissions and long-range transport really contribute that much to the large changes seen over Asia between 2018-2020 in MODIS? Is 3 years two short a window for comparison then?

Yes, exactly, we only have a shorter period of time in last panel, making the influence of natural year-to-year variability in e.g. natural sources and transport potentially more important, and hence the role of continued changes in anthropogenic emissions less clear. To try to clarify we have modified to:

*"However, we note that shorter-term variability in natural emissions and long-range transport may factor into the observed trend as well, complicating the comparison."*

Line 384 – Instead of the global evaluation plots could you also show the AOD evaluation across regions as it would highlight different areas of change and where the change in emissions has caused differences. Also could errors is aerosol process representation also be contributing to these underestimations? I am not sure it is all due to emissions.

Good point, thanks, such processes are of course also important for the simulated aerosol distributions and magnitude. We have added:

*"Other factors include the representation of processes related to aerosol transport and scavenging."*

Line 416 – "easter" should be 'Eastern'

Corrected, thanks.

Figure 2 – Need to state what year these differences are for.

Corrected, thanks.

Figure 3 – Can you make the linear trend lines a bit more distinct? Also panels in c) are currently labelled as b)

Apologies for the wrong labeling. We have modified this figure following comments from referee #1 to better highlight regional differences and have also tried to use darker/thicker lines for the trends. Hopefully this has improved the readability.

Figure 4 – can you make the colour bars wider so it easier to see the different shades

Figure has been modified and is combined with a bar chart of regional mean RF following comments by referee #1. The two bars have the same hatching, hopefully this is clearer in the new version.